



# Synergy of Using Nadir and Limb Instruments for Tropospheric Ozone Monitoring

Viktoria F. Sofieva[1], Risto Hänninen[1], Mikhail Sofiev[1], Monika Szeląg[1], Hei Shing Lee[1], Johanna Tamminen[1], Christian Retscher[2]

[1]Finnish Meteorological Institute, Helsinki, Finland
[2]ESA/ESRIN, Frascati, Italy

*Correspondence to*: Viktoria F. Sofieva (viktoria.sofieva@fmi.fi)

**Abstract.**

The satellite measurements in nadir and limb viewing geometry provide a complementary view of the atmosphere. An effective
combination of the limb and nadir measurements can provide a new information about atmospheric composition.  In this work,
we present tropospheric ozone column datasets that have been created using combination of total ozone column from OMI and
TROPOMI with stratospheric ozone column dataset from several available limb-viewing instruments (MLS, OSIRIS, MIPAS,
SCIAMACHY, OMPS-LP, GOMOS).

We have developed further the methodological aspects of assessment of tropospheric ozone using the residual method using
simulations with the chemistry-transport model SILAM. It has been shown that the accurate assessment of ozone in the upper
troposphere and the lower stratosphere (UTLS) is of high importance for detecting the ground-level ozone patterns.

The stratospheric ozone column is derived from a combination of ozone profiles from several satellite instruments in
limb-viewing geometry. We developed a method for the data homogenization, which includes the removal of biases and a-
posteriori estimation (validation) of random uncertainties, thus making the data from different instruments compatible with
each other. The high horizontal and vertical resolution dataset of ozone profiles is created via interpolation of the limb profiles
from each day to 1°x1° horizonal grid. A new kriging-type interpolation method, which takes into account data uncertainties
and the information about natural ozone variations from the SILAM-adjusted ozone field, has been developed. To mitigate the
limited accuracy and coverage of the limb profile data in the UTLS, a smooth transition to the model data is applied below the
tropopause. This allows estimation of  stratospheric ozone column with full coverage of the UTLS. The derived ozone profiles
are in very good agreement with collocated ozonesonde measurements.

The residual method was successfully applied to OMI and TROPOMI clear-sky total ozone data in combination with
the stratospheric ozone column from the high-resolution limb profile dataset. The resulting tropospheric ozone column is in
very good agreement with other satellite data. The global distributions of tropospheric ozone exhibit enhancements associated
with the regions of high tropospheric ozone production.



The main created datasets are (i) monthly 1°x1° global tropospheric ozone column dataset using OMI and limb instruments, (ii) monthly 1°x1° global tropospheric ozone column dataset using TROPOMI and limb instruments and (iii) daily 1°x1° interpolated stratospheric ozone column from limb instruments.

Other datasets, which are created as an intermediate step of creating the tropospheric ozone column data, are: (i) daily 1°x1° clear sky and total ozone column from OMI and TROPOMI (ii) Daily 1°x1° homogenized and interpolated dataset of ozone

profiles and (iii) daily 1°x1° dataset of ozone profiles from SILAM simulations with adjustment to satellite data.

These datasets can be used in various studies related to ozone distributions, variability and trends, both in the troposphere and the stratosphere.

Key words: tropospheric ozone, satellite observations, limb-viewing satellites, atmospheric composition modelling, data fusion


## 1   Introduction

The detailed information about the tropospheric ozone is of high importance because it is one of the major environmental concerns. Upper tropospheric ozone is an important greenhouse gas, which contributes to the global warming. Tropospheric ozone is also a pollutant affecting air quality. It is responsible for respiratory diseases in humans, leads to premature mortality,

and causes damage to crops and ecosystems (e.g. Jacobson, 2012; Lippmann, 1991). It was shown that the amount of tropospheric ozone increased globally during the 20th century due to enhanced emissions of anthropogenic precursors (e.g., Marenco et al., 1994; Shindell et al., 2006).

The satellite measurements in nadir and limb viewing geometry provide a complementary view of the atmosphere. These two measurement systems have their own advantages and limitations. The nadir-looking instruments have a good horizontal

resolution; they are good in retrievals of total columns, while their vertical resolution is limited. The measurements in the limb-viewing geometry have usually a good vertical resolution but their horizontal resolution is limited by the spatial sampling and cannot be better than the effective horizontal length of interaction with the atmosphere (a few hundreds of kilometers). The limb profilers allow for a good quality of trace gas retrievals in the stratosphere, while the retrievals from limb instruments in the troposphere are often problematic due to low signal-to-noise ratio and presence of clouds. An effective combination of the

limb and nadir measurements of atmospheric composition can provide a new information about atmospheric composition. Successful examples of such combination are tropospheric ozone datasets obtained by subtracting stratospheric columns from the total ozone columns, for OMI (Ozone Monitoring Instrument) nadir and MLS (Microwave Limb Sounder)  profile measurements (Ziemke et al., 2006), and for SCIAMACHY (SCanning Imaging Spectrometer for Atmospheric CHartographY)  limb-nadir matching measurements (Ebojie et al., 2016).

The retrieval of tropospheric ozone from purely nadir-looking instruments is a challenging, strongly ill-posed problem. Therefore several approaches have been developed: 1) using the spectral information in the nadir satellite measurements (nadir





profile retrievals, (Kroon et al., 2011; Liu et al., 2010a, 2010b; Mielonen et al., 2015)), 2) the convective cloud differential (CCD) method applied in the tropics (Ziemke et al., 1998), and 3) via subtraction of stratospheric column from an external source from the total ozone column (the residual method). The first study with the residual method was performed in the late

1980s by Fishman and Larsen (1987) who subtracted SAGE (Stratospheric Aerosol and Gas Experiment) stratospheric ozone from total ozone columns by TOMS (Total Ozone Mapping Spectrometer). Aside with calibration issues when using two different satellite measurements, there was also a serious constraint in producing global data with adequate temporal and spatial coverage due to sparse coverage by the SAGE solar occultation measurements. Several other residual-based approaches have been developed over the years, with combination of TOMS and MLS/UARS (Fishman et al., 1990) and OMI and MLS

(Schoeberl et al., 2007; Ziemke et al., 2006, 2011).

The main problems associated with the tropospheric ozone retrievals from nadir and limb measurements are (i) necessity of data calibration and (ii) usually insufficient horizontal coverage of limb profile measurements. In order to get the stratospheric ozone field with high-horizontal resolution, a 2D interpolation (Ziemke et al., 2006) or wind-trajectory scheme (Schoeberl et al., 2007) is used.

The satellite measurements of total ozone by TROPOMI (TROPOspheric Monitoring Instrument) on Sentinel 5P open new possibilities for monitoring of atmospheric pollutants from space because of their unprecedented horizontal resolution.

The main aim of our work is the further development of the methods for assessment of tropospheric ozone using synergy of limb and nadir measurements and applying them to measurements by TROPOMI/Sentinel 5P and OMI/Aura. The novelty of the approach is in combination of the measurements from several satellite instruments in limb-viewing geometry for the

stratospheric ozone column dataset. In addition, we have performed extensive sensitivity studies using the simulations with the chemistry-transport model (CTM) SILAM (System for Integrated modeLling of Atmospheric coMposition, Sofiev et al., 2015b, 2020).

This paper presents the description of the methods developed within the ESA project SUNLIT (Synergy of Using Nadir and Limb Instruments for Tropospheric ozone monitoring) and shows some illustrative examples of the created datasets. The paper

is organized as follows. Section 2 describes the satellite datasets and the CTM SILAM. Section 3 is dedicated to feasibility studies on retrievals of tropospheric ozone by the residual method, which have been performed using simulations with SILAM. Section 4 describes the retrieval method for tropospheric ozone column developed in the SUNLIT project. Examples of data and some validation results are shown in Section 5. Summary (Section 6) concludes the paper. Additional illustrations are provided in the Supplement.






## 2 Data and the chemistry-transport model

### 2.1 Total ozone column from nadir satellite instruments

In our analyses we use total column ozone data from OMI on Aura (https://aura.gsfc.nasa.gov/omi.html, last access 3.09.2021 Levelt et al., 2018) and TROPOMI on Sentinel 5P (http://www.tropomi.eu, last access 3.09.2021;

https://sentinel.esa.int/web/sentinel/missions/sentinel-5p, last access 3.09.2021, Veefkind et al., 2012). OMI and TROPOMI are in sun-synchronous orbits and provide the information at about same local time (1:30 p.m. and 1:45 p.m.). OMI has been operating since 2004, and its data have been used in different applications including evaluation of trends (Levelt et al., 2018). The OMI ground-pixel size is 13x25 km$^2$. TROPOMI has been operating since 2017. It has a very fine spatial resolution with the ground-pixel size 3.5x7 km$^2$ before August 2019 and 3.5x5.5 km$^2$ afterwards.

In our work, we use the Level 2 OMI and TROPOMI total ozone columns retrieved with the same GODFIT v4.0 processor (Lerot et al., 2014). Total ozone columns are derived using a non-linear minimization procedure of the differences between measured and modelled sun-normalized radiances in the ozone Huggins bands (fitting window: 325-335 nm). The typical random uncertainties of total column data, as estimated by the retrieval algorithm, are in the range of 0.5 - 5 DU for OMI and 0.5 - 2 DU for TROPOMI (Lerot et al., 2014; Sofieva et al., 2021).


### 2.2 Ozone profiles from limb satellite measurements

In our work, we use the data from several limb /occultation satellite instruments. Three of them - MIPAS (Michelson Interferometer for Passive Atmospheric Sounding), SCIAMACHY and GOMOS (Global Ozone Monitoring by occultation of Stars) - operated on Envisat (Environmental Satellite) in 2002-2012. Three other limb instruments are still operational: OSIRIS

(Optical Spectrograph and InfraRed Imaging System) on Odin, MLS on Aura and OMPS-LP (Ozone Mapping and Profiles Suite - Limb Profiler) on Suomi-NPP.

The information about the ozone profile data is collected in Table 1. All these satellites are in sun-synchronous orbits, so that the measurements are performed in nearly the same local overpass time, which is instrument-specific. MLS and OMPS measurements are performed in local times close to OMI and TROPOMI measurements, which is advantageous for the

proposed application. The abovementioned limb instruments provide ozone profiles with a vertical resolution of 2-4 km and random uncertainties 1-10 % in the stratosphere (see Table 1 for more details). The horizontal resolution associated with the limb-profile measurement technique is 200-400 km along line of sight. The selected limb instruments provide from ~100 to ~3500 profiles per day (Table 1), which are spaced uniformly in longitudinal direction according to satellite orbits (typical daily sampling patterns are illustrated in Figure 7 below).


*Table 1. Information about the datasets used in the analyses*

| Instrument/ satellite | Processor, data source | Time period | Local time | Estimated precision | Profiles per day |
|---|---|---|---|---|---|
| OSIRIS/ Odin | USask v5.10 | 2011 – present | 6 a.m., 6 p.m. | 2-10% | ~250 |
| GOMOS/ Envisat | ALGOM2s v1.0 | 2002 – 2011 | 10 p.m. | 0.5–5 % | ~110 |
| MIPAS/ Envisat | KIT/IAA V7R_O3_240 | 2005 – 2012 | 10 p.m., 10 a.m. | 1–4% | ~1000 |
| SCIAMACHY/ Envisat | UBr v3.5, | 2002- 2012 | 10 a.m. | 1-7% | ~1300 |
| OMPS/ Suomi NPP | USask 2D v1.1.0, | 2012-present | 1:30 p.m. | 2-10% | ~1600 |
| MLS/Aura | NASA v. 4.2 | 2004-present | 1:45 a.m. and p.m. | 1-7 % | ~3500 |


The accuracy and data coverage are lower in the upper troposphere and the lower stratosphere (UTLS) than in the middle stratosphere (Figure 1). For limb-viewing satellite measurements, retrievals in the UTLS are challenging due to presence of clouds and lower signal-to-noise ratio. The average estimated random uncertainties are in the range 5-30 %. Not all ozone profiles cover fully the UTLS region (Figure 1, right).

For all limb instruments, we use the ozone profiles from the HARMonized dataset of Ozone profiles (HARMOZ) developed in the ESA Ozone_cci project (Sofieva et al., 2013, https://climate.esa.int/en/projects/ozone/data/, last access 17.05.2021). HARMOZ consists of the original retrieved ozone profiles from each instrument, which are screened for invalid data by the instrument experts and are presented on a common vertical grid (the altitude-gridded profiles are used in our paper) and in a common netCDF4 format. The detailed information about the original datasets can be found in (Sofieva et al., 2013).


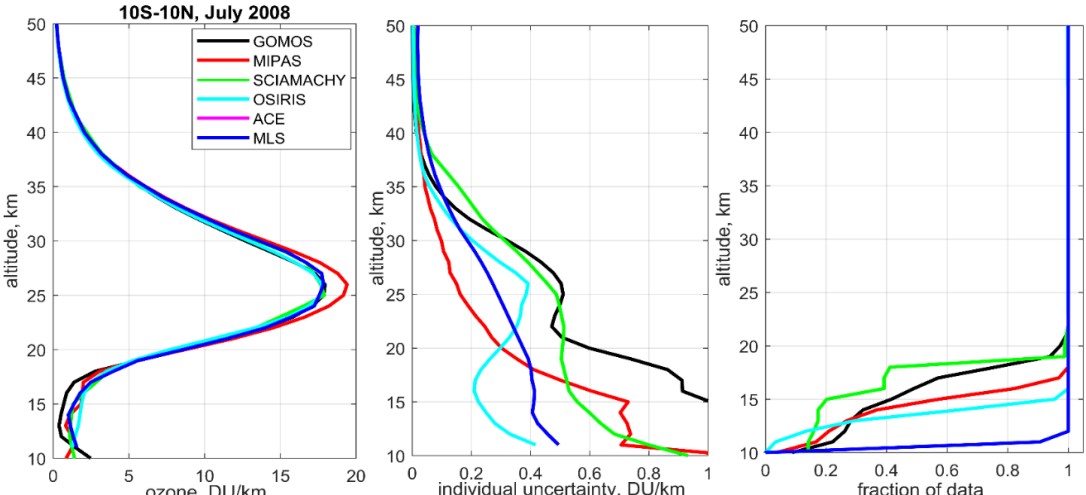

**Figure 1. Left: mean ozone profiles at 10S-10N in July 2008. Center: uncertainty of individual ozone profiles, right: fraction of data (with respect to the data in the stratosphere).**



## 2.3  SILAM chemistry-transport model

The modelling tool used in the project is the System for Integrated modeLling of Atmospheric coMposition SILAM (Sofiev et al., 2015, http://silam.fmi.fi, last access 3.09.2021). This is an offline chemistry-transport model that has several unique features making it highly suitable for the current application. SILAM is a multi-scale model with seamless scaling from the global coverage down to regional scale with 1-km resolution (Korhonen et al., 2019; Kouznetsov et al., 2020; Sofiev et al., 2018; Xian et al., 2019). SILAM chemical and physical modules cover both the troposphere and the stratosphere (Hänninen et al., 2020; Kouznetsov and Sofiev, 2012; Sofiev, 2002; Sofiev et al., 2020).

SILAM is an extensively evaluated model, a member of the Copernicus Atmospheric Monitoring Service (CAMS) regional European ensemble (https://www.regional.atmosphere.copernicus.eu/, last access 3.09.2021)  and the Panda-MarcoPolo (an EU FP7 project) ensemble for Asia, both operational services with established daily evaluation procedure (Brasseur et al., 2019; Kukkonen et al., 2012; Petersen et al., 2019; Xian et al., 2019). The model has also extended data assimilation capabilities (Sofiev, 2019; Vira et al., 2017; Vira and Sofiev, 2012, 2015).

In this work, we used the ozone profiles simulated with new development of SILAM v5.7 with the horizontal resolution 1°x1° and the vertical grid as in the ERA-Interim dataset. Compared to v5.6, the new SILAM version has improved photolysis rates, advanced characterization of the clouds and aerosols effects, together with dry and wet deposition where the scavenging has separate parameters for ice and water clouds. For meteorological parameters, we have moved from ERA-Interim to new ERA5 data set which has hourly time resolution. In addition, the newly implemented CBM05 (Carbon Bond Model from year 2005) chemistry module (https://camx-wp.azurewebsites.net/Files/CB05_Final_Report_120805.pdf, last access 3.09.2021) provides better tropospheric ozone concentrations, especially in tropics and in remote regions, compared with the previous CBM4 chemistry (Carbon Bond Model version 4, (Gery et al., 1989) and updates).

For anthropogenic emissions, we use CAMS (Copernicus Atmosphere Monitoring Service) global emission database (v2.1) together with EDGAR4.3.2 emissions for aviation and partly self-made emissions for most important CFC-compounds. In addition, SILAM takes into account biogenic emissions of isoprene and monoterpene (database based on the MEGAN model), sea-salt emissions (including its small bromine fraction), dust-emissions, and NOx emissions from lightning. Emissions from fires are also included, either using IS4FIRES (see: http://silam.fmi.fi/fires.html) or other emission inventories.

For the majority of analyses presented in this paper, daily averaged ozone fields are used.

## 3  Feasibility studies on residual method to retrieve tropospheric ozone

## 3.1  Tropospheric ozone features observable by the residual method

About 90% of ozone is in the stratosphere (the ozone layer). Figure 2 shows a typical ozone profile for the equatorial region, in units of DU/km, with indicated contributions from different layers. The challenges associated with the residual



method are clearly seen in Figure 2: the ozone in the UTLS has nearly the same abundance as the lower tropospheric ozone, both much smaller than the stratospheric ozone column.

The ozone enhancement in the troposphere at the altitudes below 5 km is a result of complicated interplays of chemical production and loss mechanisms controlled by the abundance of the key chemical agents (NOx and volatile organics), environmental conditions (solar radiation and temperature), and surface uptake by vegetation. Revealing the resulting patterns

is therefore a challenging task, especially because these high-frequency spatial and temporal fluctuations have to be distinguished from the fluctuations in the ozone layer and noise in the limb observations, which can be comparable with the tropospheric signal itself.

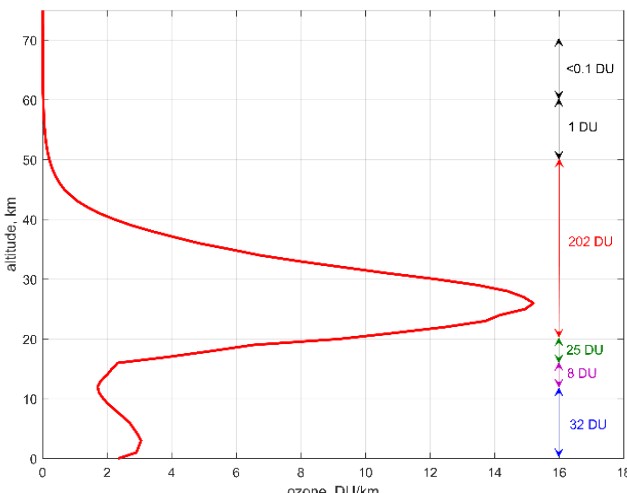

**Figure 2. Typical ozone profile for the equatorial region, with indicated contributions of different layers to the total ozone column.**

To facilitate the development of the residual method and find the best feasible spatio-temporal resolution for the dataset, we have performed feasibility analyses with the SILAM CTM. The model data is either used in their entirety or sub-sampled at locations and times of satellite measurements.

Throughout this paper, the thermal tropopause definition is used to distinguish between the troposphere and the

stratosphere (WMO, 1957). In some special cases at high latitudes, when this definition fails to find the tropopause, we use an ozonepause defined as the altitude where the ozone concentration drops (looking from the stratosphere) down to 3.5 DU/km.

### 3.2   The effect of vertical integration

When considering the tropospheric ozone column, it is expected that the ground-level ozone enhancements will be clearly visible but smeared out and displaced due to advection. This feature is illustrated in Figure 3, which compares the

ground-level SILAM ozone data (Figure 3, left panels) with the tropospheric ozone columns reaching from the ground, either





up to 3 km below the tropopause, or up to the tropopause (Figure 3, center and right panels, respectively) for 1 July 2008 (upper rows) and averaged over the whole month (bottom panels).

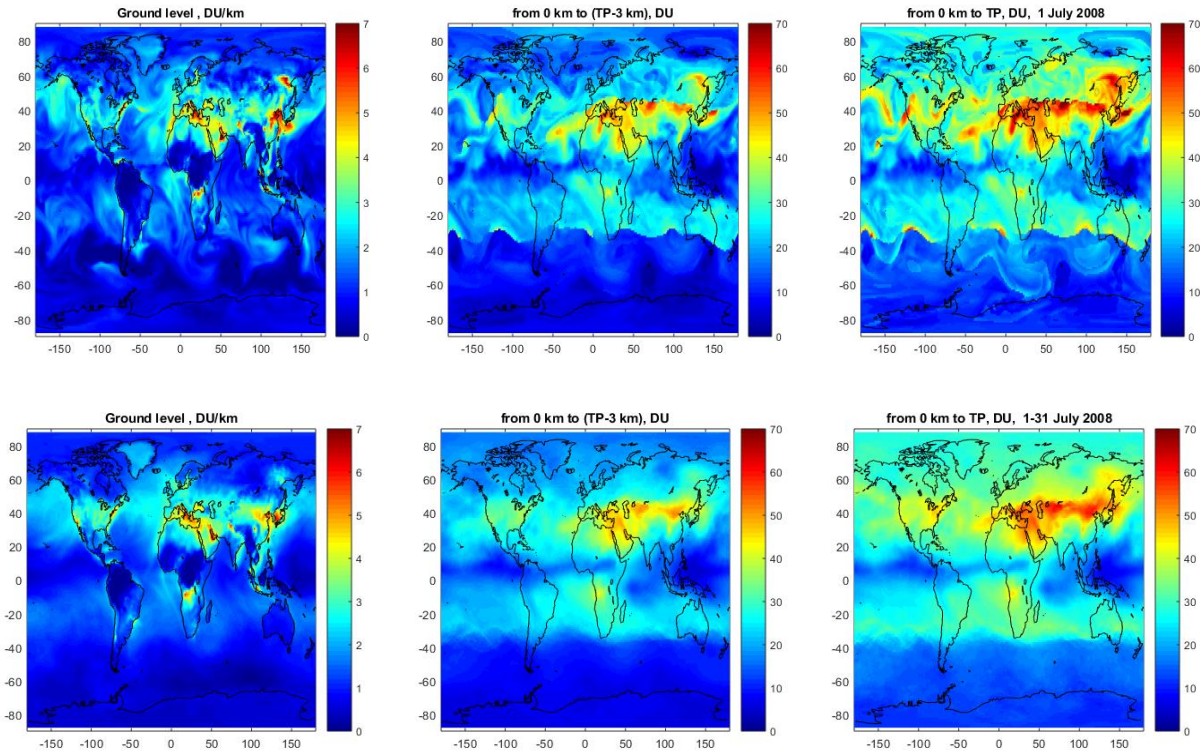


**Figure 3. Simulations with SILAM for 1 July 2008 (top panels) and monthly mean for July 2008 (bottom panels). Left: ground level ozone, center: truncated tropospheric ozone column (from ground to the altitude 3 km lower than the tropopause, right: the full tropospheric ozone column (from ground to the tropopause).**

As seen from Figure 3, tropospheric ozone column (integrated from the surface to the tropopause and referred hereafter to as full tropospheric ozone column, "full TrOC", Figure 3, right panels) has a large portion of ozone from the UTLS region, so that the tropospheric features (especially close to the ground level) are significantly blurred in the tropospheric column. If we consider the altitude range from the ground to 3 km below the thermal tropopause (referred to as "truncated TrOC", Figure 3 center panels), the influence of the UTLS is smaller but still significant.

In the monthly averaged fields (Figure 3, bottom panels), the ground level ozone enhancements are visible but smoothed. The choice of the upper limit of the tropospheric ozone integration (up to tropopause or below, compared central and right panels in Figure 3) influences the overall level of tropospheric ozone column (as expected) and also the contrast of



local enhancements. The higher contrast of the details visible from the truncated TrOC is advantageous for detecting the lower tropospheric structures.

Since the quality of limb-profile data (both accuracy and coverage) in the UTLS is limited, one can consider possibility of estimating the upper tropospheric ozone (for example, the layer of 3 km below the tropopause) and subtracting it from the full TrOC (analogy of ghost column correction in retrievals from nadir-looking instruments). To illustrate the effect, we simulated two approximate corrections of the upper tropospheric ozone. In the first correction, the upper-troposphere (UT, from 3-km below the tropopause up to the tropopause) monthly zonal mean ozone column was computed from the SILAM

data, for each latitude zone and subtracted from each data point of full TrOC, for each day. In the second correction, the UT ozone column correction is done using the tropopause-related ozone climatology TpO3 (Sofieva et al., 2014). We found that even such very approximate upper tropospheric ozone corrections give the monthly map of truncated TrOC nearly identical to the true one (Figure S1, right panels), with the difference to the true values mostly smaller than ±3 DU.

### 3.3     The effect of sampling and averaging kernel

The daily horizontal coverage by limb instruments is limited (see examples in Sect. 4.3). If the monthly mean stratospheric ozone column (SOC) is computed via simple averaging the data with such sampling, the resulting SOC has significant deviations from the SOC computed using the full ozone field, because different pixels are covered by data from different days. The approach of averaging first the stratospheric ozone column and then subtracting it from the averaged total ozone column produce pronounced errors, due to limited sampling by limb instruments (illustration can be found in the Supplement, Figure

S2). This implies that the monthly average of tropospheric ozone column should be constructed from its daily values.

Tropospheric ozone column computed via averaging daily TrOC obtained by the residual method is quite close to the true distribution using the data with full coverage. This is illustrated in left and central panels of Figure 4. The right panel of Figure 4 shows an analogous estimate of the tropospheric ozone, in which the total ozone column was computed with the OMI averaging kernels taken into account (the examples of OMI and TROPOMI averaging kernels are shown in supplementary

Figure S3). Since OMI and TROPOMI are sensitive to middle and upper tropospheric ozone (Figure S3), the tropospheric ozone column derived by the residual method also misses a substantial fraction of the near-surface pattern. An interesting feature, which is associated with the influence of averaging kernel, is that the enhancements over central Africa are shifted to Atlantic Ocean. This is a combined effect of OMI low sensitivity near the ground and wind advection of both ozone and its precursors towards the west in the middle troposphere.


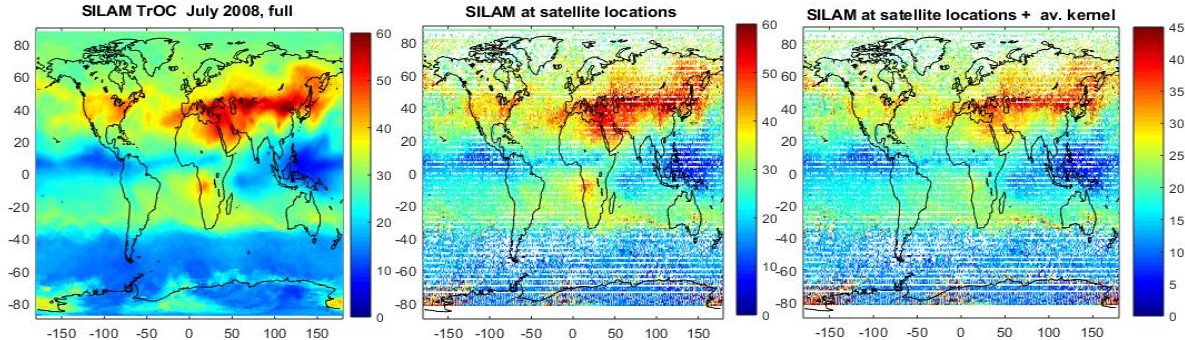

**Figure 4. Estimates of full tropospheric ozone column (from the ground to the tropopause) using the application of the residual method to SILAM ozone profiles, monthly average for July 2008. Left: TrOC using the full SILAM ozone field, center: SILAM data sub-sampled at locations of nadir and limb satellite instruments. Right: averaging kernel is taken into account in computing total ozone column at OMI locations.**

### 3.4    Conclusions from feasibility studies on the residual method

The following main conclusions can be drawn from the feasibility studies:

- In order to detect ground enhancements of tropospheric ozone, both the stratospheric and the UTLS contribution should be accurately removed from the nadir total ozone column data, because the UTLS ozone contribution is comparable with the lower-tropospheric ozone abundances and the stratospheric one largely exceeds it.

- The sensitivity of nadir-looking satellite instruments is limited in the lower troposphere, therefore the observed ground-level ozone enhancements are shifted from the near-surface production areas and blurred, as a consequence of atmospheric motions and chemical transformations.

- Due to large variability of ozone field and limited sampling by satellite instruments, monthly average tropospheric ozone column from combination of nadir and limb instruments, should be computed from daily tropospheric ozone column.

- Upper tropospheric ozone column correction using the data from an external source is an attractive approach, which allows removal of the UT contribution from the full tropospheric ozone column without introducing large uncertainty into the truncated tropospheric ozone column.

Based on these studies, we have developed the method of estimating the tropospheric ozone column using the combination of limb and nadir measurements. The specific feature of our method is using the CTM-simulated ozone field in creating high-spatial-resolution ozone field, in the stratosphere and the UTLS.

In the next section we present the detailed description of the retrieval algorithms.






# 4  Tropospheric ozone column by the residual method

## 4.1  Methodology in general

We follow the general idea of the residual method, which consists of (1) creating a clear-sky total ozone column from nadir instruments, (2) creating a high-horizontal resolution stratospheric ozone column by combining ozone profiles from several 265 limb instruments, and (3) evaluating the tropospheric ozone column as the difference between the total and the stratospheric columns. The computations are done at daily level making the tropospheric ozone columns with 1°x1° spatial resolution, which are subsequently combined to the monthly mean column.

## 4.2  Gridded datasets from nadir instruments

To create daily gridded total ozone column in 1°x1°latitude-longitude bins (which are often referred to as Level 3), we 270 used the clear sky Level 2 data, with cloud fraction less than 0.2.

Since the OMI row anomaly is not fully characterized by the processing flags, an additional adaptive data filtering was applied. First, we removed flagged pixels and one additional row from each side of the flagged region. The presence of row anomaly was also checked by evaluating the ozone difference in adjacent rows. If the values higher than 100 DU are detected, the whole region between the two "ozone jumps" is removed. Finally, only the data with relative uncertainty less than 4% are 275 used for creating the daily gridded data.

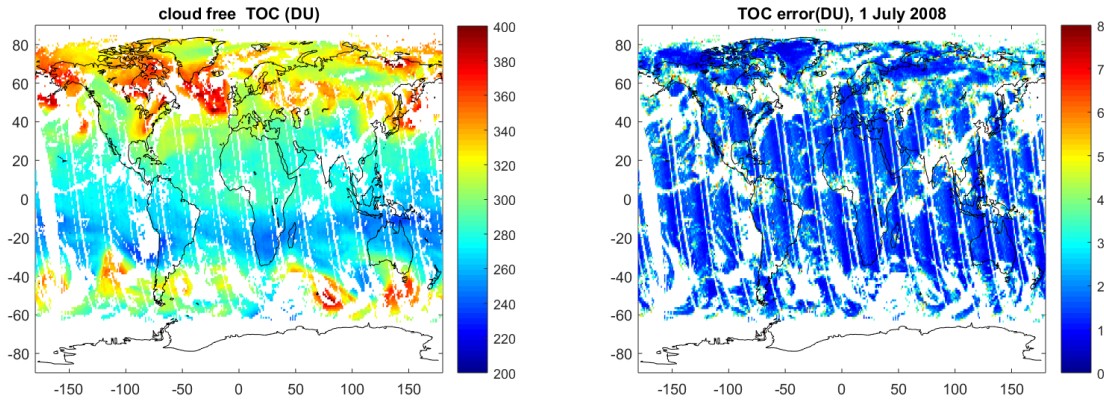

**Figure 5. OMI Level 3 total ozone column data (left) and random uncertainty (right) for 1 July 2008.**

In each latitude-longitude bin, the mean of total ozone column data is evaluated. The uncertainty of the total ozone column is computed as:

$$\sigma^2 = \frac{1}{N}\sum_i \sigma_i^2 + \frac{1}{N}\operatorname{var}(\rho_i) \qquad (1)$$





where $\sigma_i$ are uncertainties reported by the retrieval algorithm and $\mathrm{var}(\rho_i)$ is the variance of N individual ozone values in the bin. The typical daily gridded clear-sky total ozone column and the corresponding random uncertainties are shown in

Figure 5.

The daily average gridded TROPOMI total ozone column data are computed in a similar way, with the same spatial resolution 1°x1°.

### 4.3    Homogenized and interpolated dataset of ozone profiles

In our approach, we first create the 1°x1° gridded and interpolated dataset of ozone profiles, and then we compute

stratospheric column via integration of ozone profiles. We selected such approach because the limb instruments have limited accuracy and highly non-uniform coverage in the UTLS, while the accurate knowledge of the UTLS profiles is essential for application of the residual method.

In our algorithm, the creation of homogenized interpolated dataset of ozone profiles consists of three main steps:

(1) Homogenization of ozone profile data from the limb satellites measurements;

(2) Interpolation of the limb profiles from each day to 1°x1° horizontal grid;

(3) A smooth transition to the adjusted model data below the tropopause.

Below we present the detailed description of the processing.

### 4.3.1    Homogenization of ozone profile data from the limb instruments

For horizontal interpolation, the data from different satellite measurements need to be compatible. As the first step of

such data homogenization, biases between datasets are removed.

We use MLS as a reference dataset. For all other instruments, the biases with respect to MLS are evaluated for each month and for each latitude (with 1° increment), using 10° overlapping zones and corrected via adding latitude-dependent offset. This procedure removes the biases between the limb datasets, as illustrated in Figure 6. After the bias correction, the data from different instruments can be used together. An example of bias-corrected data is shown also in Figure 7 (left panel).

The optimal implementation of the horizontal interpolation method (see Sect. 4.3.2 for details) requires that the error estimates from different instruments agree and realistically describe the variations caused by random data uncertainties. However, this is not the case for the considered limb instruments: while biases between the instruments are rather small (within 10%), the estimated uncertainties can differ by an order of magnitude. This is illustrated in Figure 7, which show ozone and the reported uncertainties at 10 hPa for MLS, OSIRIS and OMPS-LP (processed by University of Saskatchewan v1.1.0).

Uncertainty estimates of OMPS data processed by University of Bremen have smaller difference with respect to MLS, but they still can differ by the factor of 2-3. The difference in error estimates depends on latitude, altitude, and season.





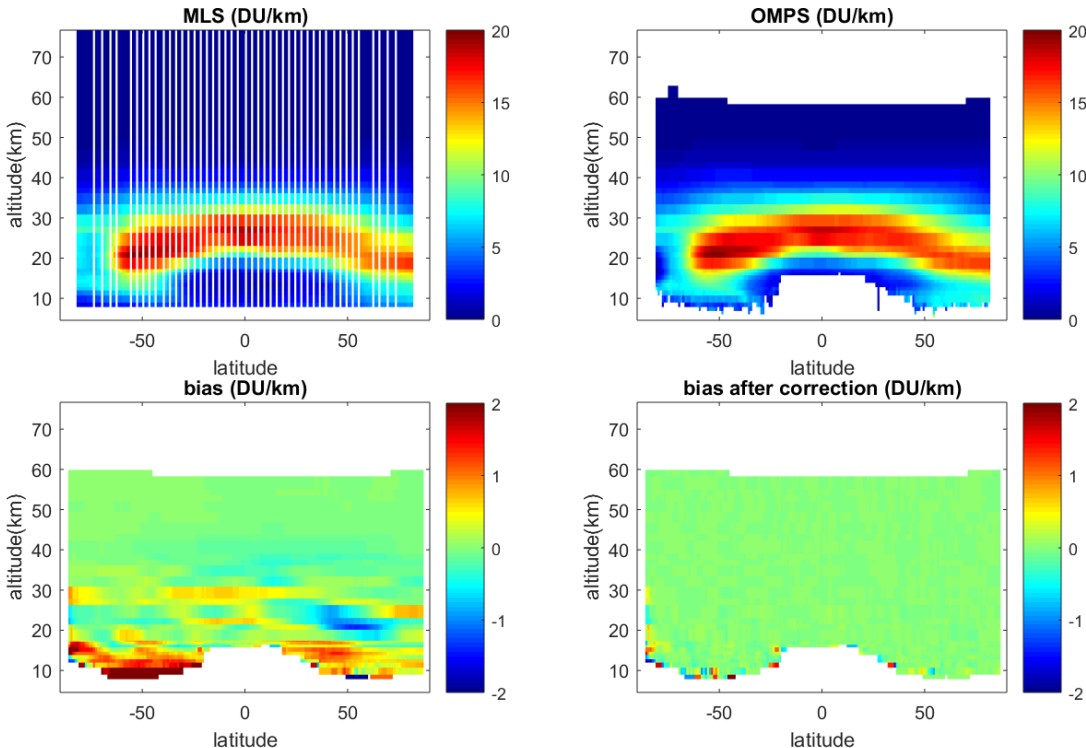

**Figure 6. Illustration of the bias correction for September 2018. Upper panels: MLS and OMPS profiles averaged over latitude zones and over the month. Bottom panels: estimated biases before (left) and after (right) bias correction.**

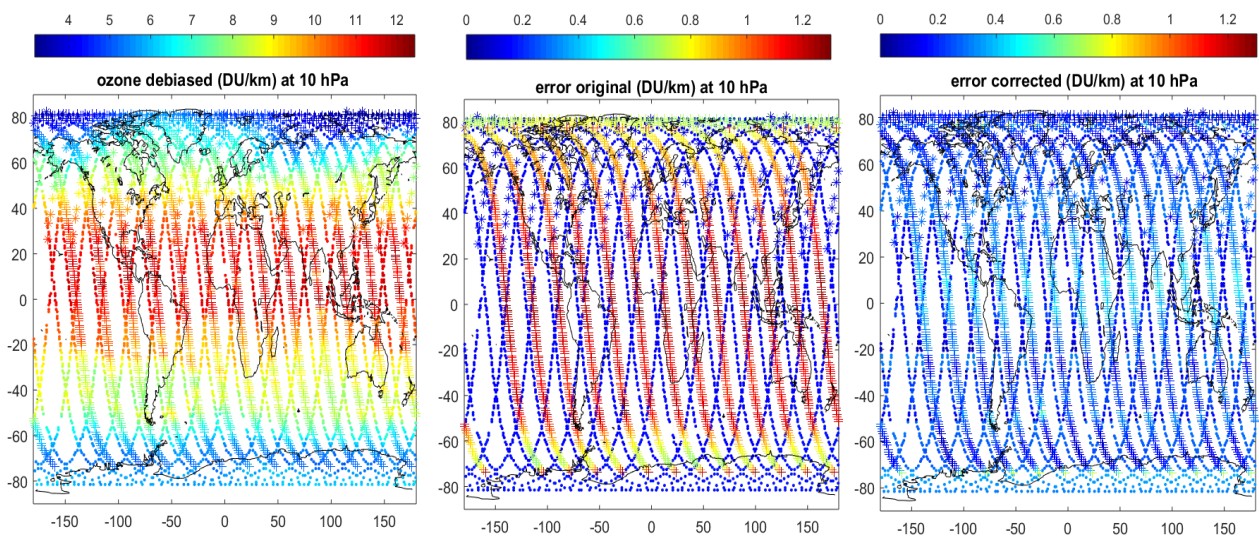

**Figure 7. Debiased ozone at 10 hPa for 1 Sep 2018 (left), corresponding original uncertainties (center), and corrected uncertainties (right). MLS data are indicated by dots, OSIRIS - by stars and OMPS by plusses.**






Therefore, we applied a simple approach that provides the random uncertainties that are consistent with the variability field. For each instrument and each month, we evaluated sample variance $s^2$ in 10° latitude zones from experimental data and the SILAM-adjusted field, which is sub-sampled at measurements locations. The adjustment of SILAM data to MLS measurements is described in the Supplement, Sect. S2. This sample variance $s^2$ provides the estimated of the natural

variability $\sigma_{nat}^2$. Then a posteriori (ex-post in von Clarmann et al. (2020) terminology) uncertainties can be estimated as

$\sigma_{ex-post}^2 = s^2 - \sigma_{nat}^2$ We computed latitude and altitude dependent offset $\Delta = \sigma_{ex-post} - \sigma_{ex-ante}$ ($\sigma_{ex-ante}$ is the mean error estimate provided with profiles, in the same 10° latitude zones over the month ) and applied it to each profile. As shown in the right panel of Figure 7, this simple correction of the uncertainty estimates makes them comparable. By the construction, they are also compatible with the observed ozone variability.

**4.3.2    Interpolation of the limb profiles**

After homogenization, the limb data are interpolated to form a high-spatial resolution dataset. For our application, the most attractive approach is a kriging-type interpolation, in which both data uncertainty and the structure of the data variability are taken into account. In this approach, the value at the point $\mathbf{r}$ is taken as a weighted mean of data in the neighbourhood:

$$x(\mathbf{r}) = \sum_i w_i x(\mathbf{r}_i),\tag{2}$$

with the weights $w_i$ inversely proportional to the total uncertainties:

$$\sigma_{tot,i}^2 = \sigma_{noise,i}^2 + D(\mathbf{r}_i - \mathbf{r}),\tag{3}$$

where $\sigma_{noise}^2$ is the estimate of the noise in the data, and $D(\mathbf{r}_i - \mathbf{r})$ is the uncertainty due to the spatial mismatch, which is usually estimated via the structure function. The structure functions are widely used in studies of small-scale natural variability and they can be used also for validation of random data uncertainties (Sofieva et al., 2021 and references therein). The

evaluation of the structure functions is discussed also in the Supplement, Sect S3. In our interpolation method, $D(\mathbf{r}_i - \mathbf{r})$ is taken from the adjusted SILAM field, for each day. The weighted mean is assessed using the 10°x20° latitude -longitude area around each point.

We have tested our interpolation scheme on the noise-free and the noisy simulated data with SILAM and found that the kriging-type interpolation described above is superior to the triangulation-type interpolation (for example, natural

neighbour interpolation, Sibson et al., 1991): the interpolation error is smaller and fine structures are better resolved. For noisy simulated data, the interpolation error is the smallest if the uncertainty estimates in Eq. (3) are realistic, as expected.



The interpolation of ozone profiles is performed at each pressure level separately. The example of the interpolated field is shown in Figure 8.

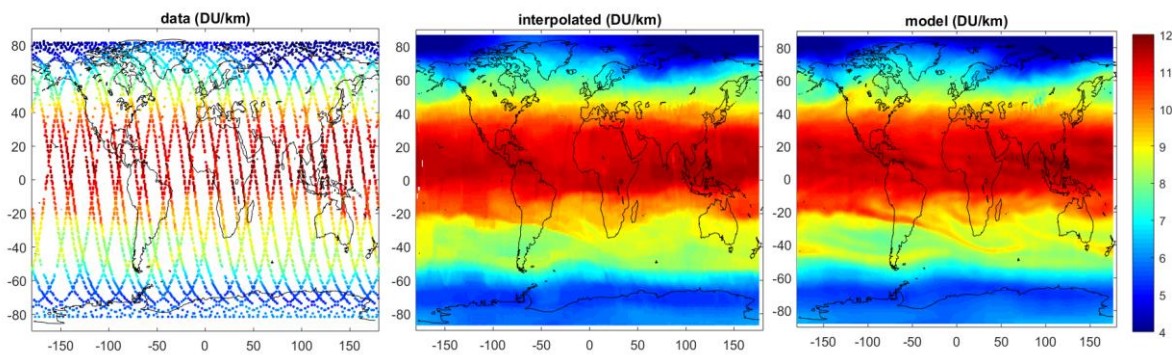


**Figure 8. Left: ozone at limb satellite measurements at 10 hPa on 1 September 2018. Center: after interpolation. Right: corresponding adjusted SILAM ozone field at the same pressure level.**

The uncertainties of the interpolated field are estimated as follows. The uncertainty after the kriging is estimated as the minimal

value of $\sigma_{tot}$ (Eq.(3)) in the bin used for weighted mean. In addition, we estimated the interpolation uncertainty using the SILAM data: we applied the same interpolation on SILAM ozone sub-sampled at measurements locations and evaluated the error as the absolute difference of true and interpolated data. The final uncertainty is the root-mean-square of error propagation and model-assessed interpolation errors. The uncertainty estimation in the interpolated ozone field is illustrated in Supplement, Sect. S4.

**4.3.3    Extension into the troposphere**

Since satellite data have limited accuracy, non-homogeneous and rather sparse coverage below the tropopause, we extended the satellite-based ozone profiles to lower altitudes by using the smooth transition to the adjusted SILAM profiles. The linear transition is performed in such a way that above 200 hPa the profile follows fully the experimental data and below 400 hPa - fully the model data. The illustration of the transition to the model data at lower altitudes is shown in Figure 9, for

tropical and polar atmospheres.

Below in Sect.5, we show that the resulting ozone profiles are in a good agreement with ozonesonde data.





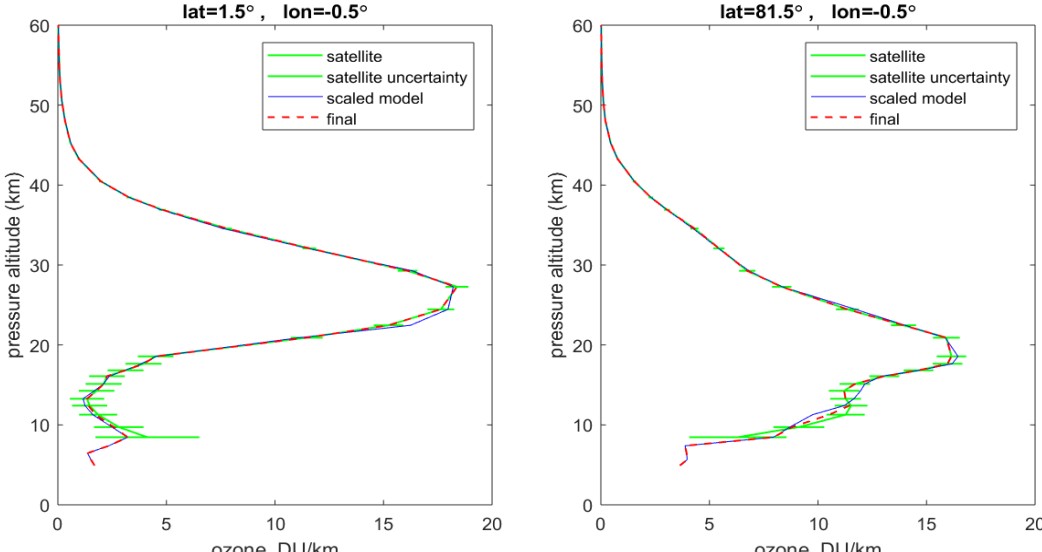

**Figure 9. Illustration of transition to model-adjusted profiles at lower altitudes for tropical (left) and polar region (right). The vertical coordinate is pressure altitude** $z = 16 \log_{10}(P_0 / P)$**, where** $P_0$**=1013 hPa is the standard pressure and** $P$ **is pressure in hPa.**

## 4.4 Stratospheric ozone column dataset

Computing the stratospheric ozone column from the high-resolution profiles is rather straightforward. The integration can be done from the tropopause upwards (we use 55 km as the upper integration limit), or from a certain altitude level. Relatively high vertical resolution of limb instruments (2-4 km) and good accuracy (Table 1) allow accurate determination of the stratospheric ozone column. Limb ozone profiles were interpolated to 100 m altitude grid and integrated by the trapezoidal method. The uncertainties are estimated using the error propagation. The examples of stratospheric ozone columns from the tropopause and from 3 km below the tropopause and corresponding uncertainties are shown in Figure 10. The estimated uncertainty of the derived stratospheric ozone column is mostly 5-8 DU (< 2%).



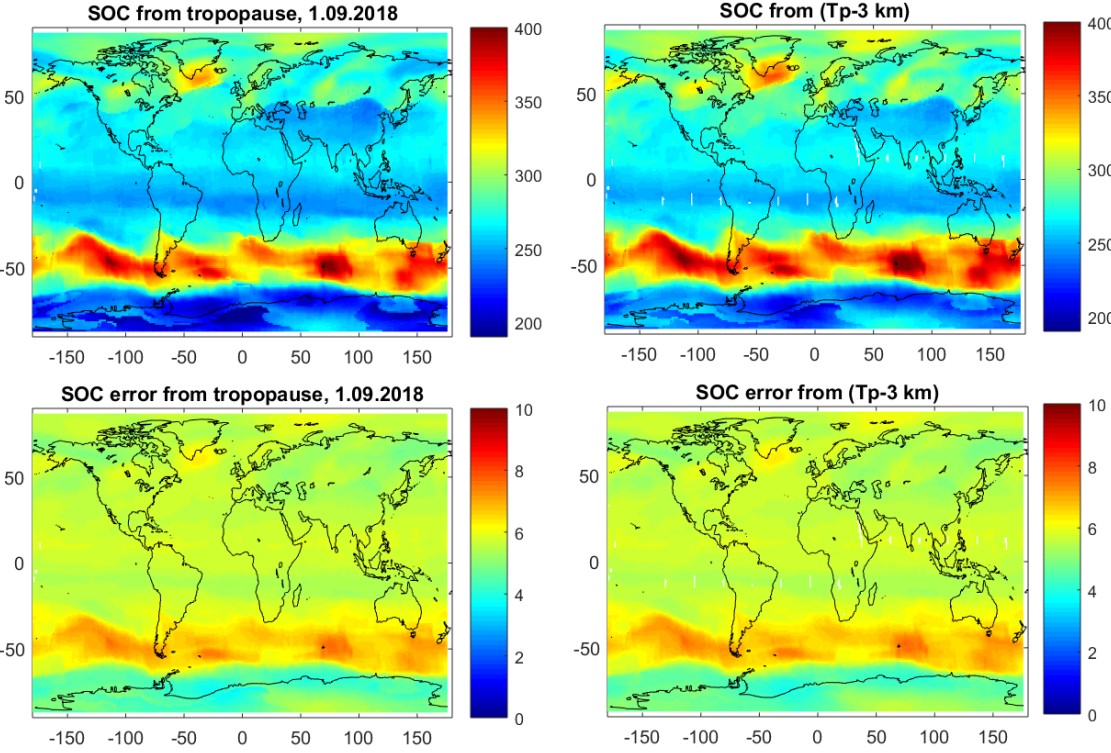

**Figure 10. Stratospheric ozone column (DU) from tropopause (left top) and from 3 km below the tropopause (right top) computed from 1°x1° merged (homogenized and interpolated) limb ozone profiles. The corresponding uncertainties are shown in bottom panels.**

## 4.5    Tropospheric ozone column

Once the high-resolution stratospheric ozone column dataset is created, the application of the residual method is straightforward: the stratospheric columns are subtracted from the clear-sky measurements by the nadir sensors, daily. The daily values can be averaged to monthly mean values subsequently.

However, before the application of the residual method, the compatibility of limb and nadir data should be checked. For this, we compared OMI and TROPOMI measurements in cloudy conditions (the ghost column is removed) with the integrated ozone profiles from the cloud-top height. For this comparison, we selected cloudy pixels with cloud fraction > 0.8 and cloud-top pressure less than 350 hPa and the corresponding limb profiles from the adjusted SILAM field. We found that over Indonesia and Western Pacific where high clouds are observed, the mean difference between nadir and limb instruments is very small, ~2 DU, for both OMI and TROPOMI. The illustrations of this comparison can be found in Supplement S5.





Although the compatibility of nadir and limb instruments in the tropics is good, there are possible data mismatches that lead to negative tropospheric ozone values at some pixels, which can be due to interpolation errors or the adjustments in the UTLS. Therefore, before averaging into the monthly mean tropospheric ozone data, we ignored the daily TrOC values smaller than -5 DU (but we allow small negative values, which can be due to noise).

Uncertainties of daily tropospheric ozone values are estimated as:

$$\sigma_{TrOC}^2 = \sigma_{TOC}^2 + \sigma_{SOC}^2 . \qquad (4)$$

The uncertainties of monthly average data are estimated similarly to uncertainties of the gridded data, i.e., by Eq.(1).

After the averaging, we performed the additional data quality control and removed unreliable data from the dataset. First, we added an offset 2 DU to the dataset, which removes the mean bias between limb and nadir stratospheric columns. Then we filtered the data with uncertainties larger than 200 % or smaller than 2 %. In the polar regions, the retrieval of tropospheric

ozone has additional challenges due to the presence and perturbations of polar vortex and loosely defined tropopause height. To exclude unrealistically large values at the latitudes larger than 65°, we filtered out those data, which are either larger than 80 DU or larger than 60 DU and with the relative uncertainty less than 10%.

The resulting tropospheric ozone distributions from OMI and TROPOMI for September 2018 are shown in Figure 11 (left panels). These distributions are very similar, but TROPOMI TrOC is less "noisy". Typical ozone enhancements for

September are observed: over Africa associated with forest fires, over China and over Mediterranean regions. Zooms on China and USA are shown in Figure 12 where one can observe the enhancements associated with large cities (but they are blurred and displaced, as expected).

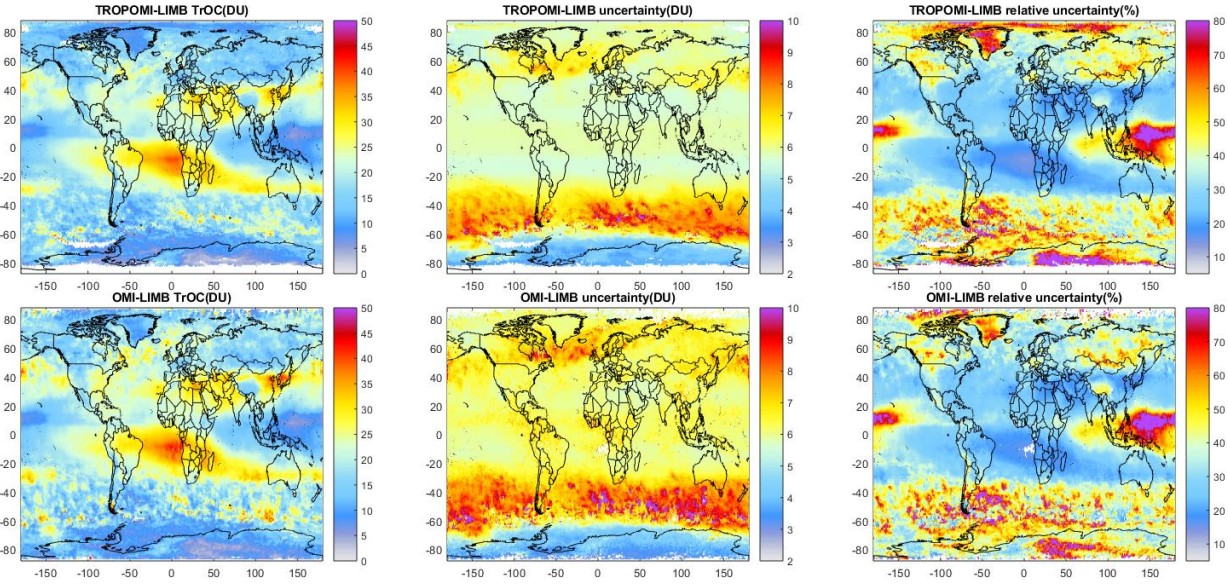

**Figure 11. SUNLIT tropospheric ozone distributions (DU, color) for September 2018, from TROPOMI (top left) and OMI (bottom**
**left). The stratospheric ozone column is estimated from 3 km below the tropopause. The corresponding estimated uncertainties are shown in absolute values (DU) on central panels and in relative values (%) on right panels.**





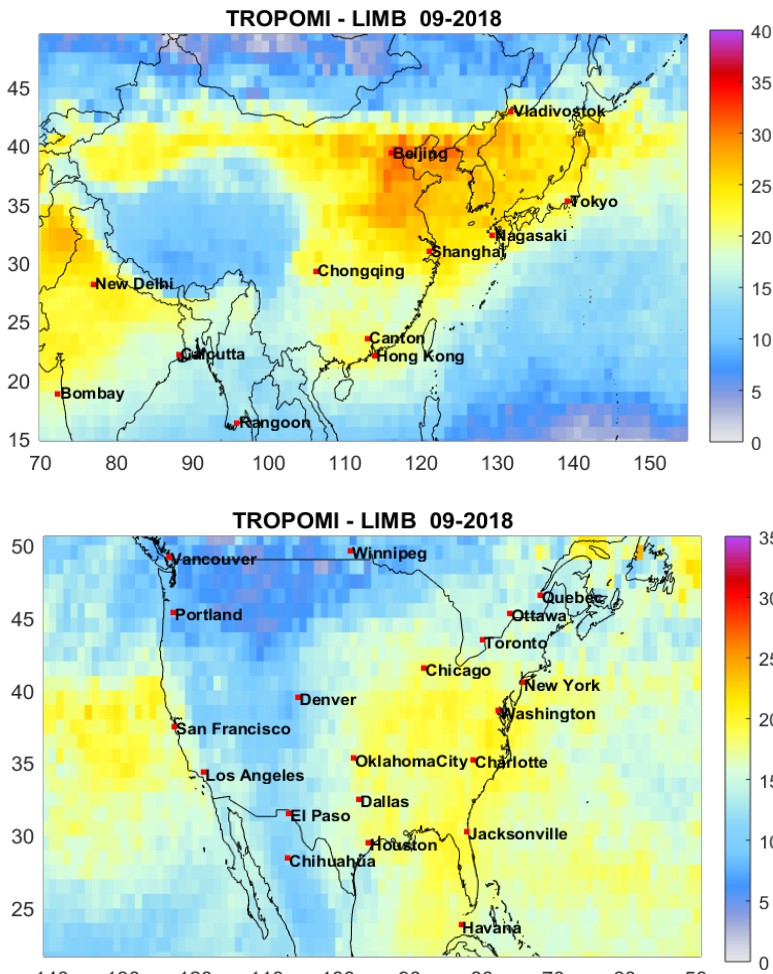

**Figure 12. As Figure 11 (top left panel), but with zoom on China (top) and USA (bottom). Color indicates tropospheric ozone column in DU.**


The estimated uncertainties for the September 2018 tropospheric ozone distributions from OMI and TROPOMI are shown in Figure 11, in absolute values (DU) on central panels and relative values (%) on right panels. The uncertainties are slightly smaller for TROPOMI than for OMI. This is due to the more accurate TROPOMI total ozone column measurements and the better coverage: due to the better horizontal resolution, it is easier to find cloud-free data in 1°x1° bins, which are used for

evaluation of tropospheric ozone column. In the majority of tropical locations, the estimated uncertainty of the tropospheric ozone column is 4-6 DU for TROPOMI and 5-7 DU for OMI. Over Indonesia, where the tropospheric ozone column has the smallest values, the relative uncertainty increases in to 100%. In the mid-latitudes of the Northern Hemisphere (summer season in August 2018), the estimated uncertainties are mostly within the range of 15-40%. The largest uncertainty is close to the polar vortex boundary, as expected.





## 5 A limited validation and examples of the data

### 5.1 Comparisons of ozone profiles with ozone sondes

To assess the quality of the high-resolution SUNLIT ozone profiles, we compared them with the ozonesonde data. For this comparison, we used the collection of ozonesonde data from the BDBP database (Hassler et al., 2008) in 2004-2006. In these comparisons, ozonesonde data are smoothed down to 1 km vertical resolution, and they are collocated with SUNLIT data within a day and 1° in latitude and longitude from the station location. The information about the selected ozonesonde data is collected in Table 2.

**Table 2. Information about ozone sonde data used in comparisons**

| Station name | Latitude | Longitude | Num. of collocations | Data sources |
|:---:|:---:|:---:|:---:|:---:|
| Eureka | 80.04 | −86.17 | 58 | WOUDC |
| Payerne | 46.49 | 6.57 | 165 | WOUDC |
| San Cristobal | −0.92 | −89.60 | 32 | WOUDC & SHADOZ |
| Irene | −25.91 | 28.21 | 32 | WOUDC |
| Neumayer | −70.65 | −8.25 | 85 | WOUDC |
| Heredia | 10.00 | −84.11 | 20 | WOUDC & SHADOZ |

Several examples - for polar, tropical and mid-latitude stations, in winter and in summer - are shown in Figure 13. As observed in this figure, ozonesonde and limb profiles are in very good agreement. The results of the statistics of differences (sonde minus satellite) for the selected stations - the median and 16th and 84th percentiles – are shown in Figure 14. The biases are small in both stratosphere and the troposphere; the inter-percentile range of differences is a few percent in the stratosphere and in the range of 10-50 % in the UTLS and the troposphere.





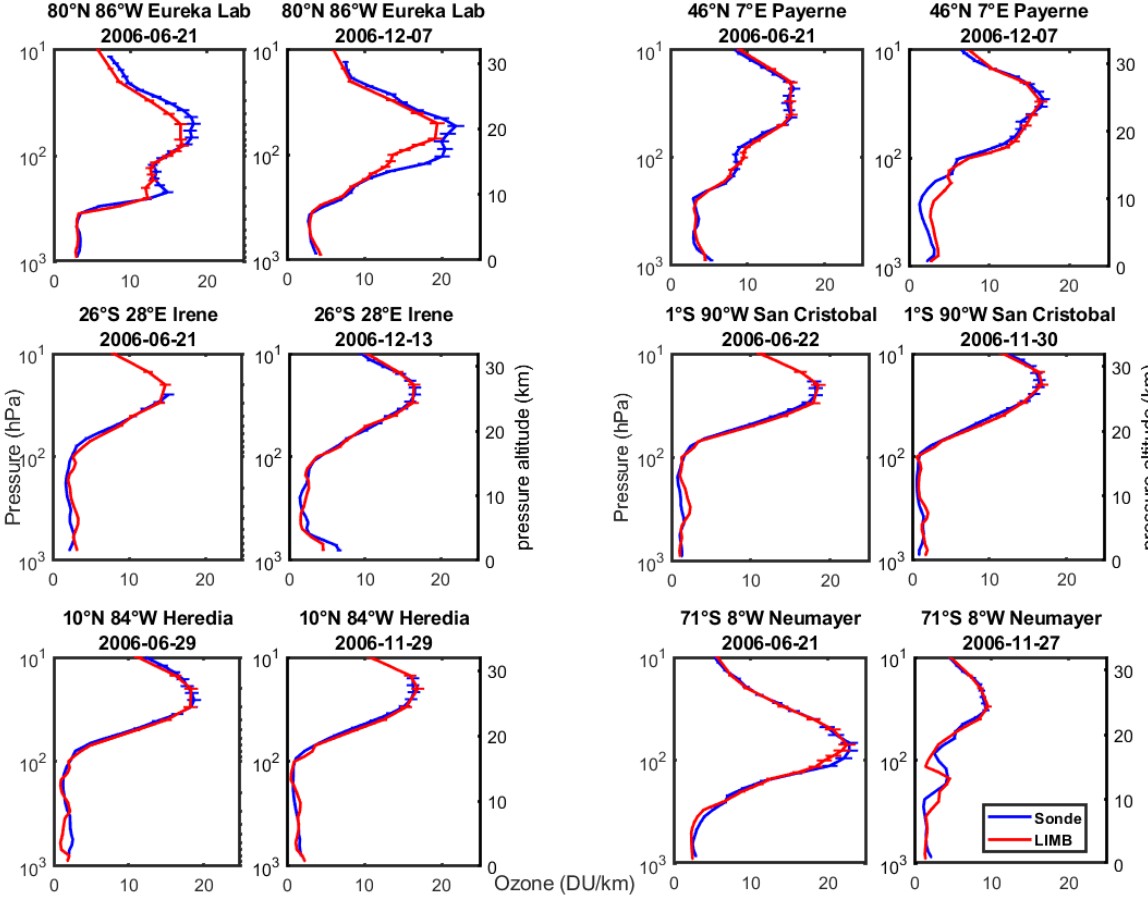

**Figure 13. Several examples of ozonesonde data (blue lines with 1-σ uncertainties) with the collocated interpolated limb profiles (red lines with 1-σ uncertainties).**





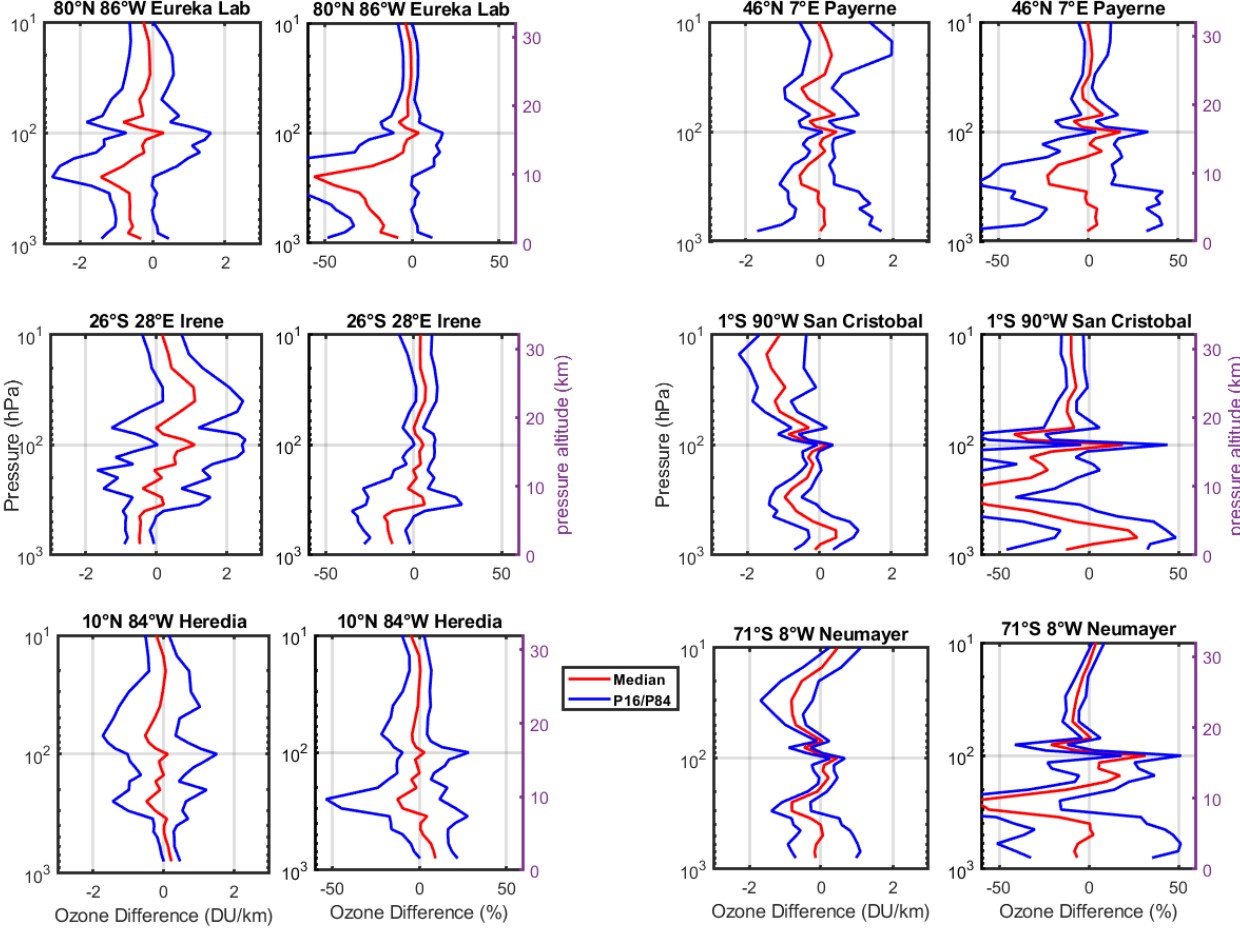

**Figure 14. The statistical parameters of differences between ozonesonde and collocated interpolated limb profiles. The red lines show the median of absolute (left panels) and relative (right panels) differences, while blue lines show the 16th and 84th percentiles.**

## 5.2    Comparison with OMI-MLS

For comparison with the NASA OMI-MLS tropospheric ozone column (obtained from https://acd-
ext.gsfc.nasa.gov/Data_services/cloud_slice/new_data.html), we computed the stratospheric ozone column from the tropopause, as it is done in the OMI-MLS dataset. The examples of tropospheric ozone column for July 2008 are shown in Figure 15. The SUNLIT tropospheric ozone distributions are provided also at high latitudes, while the OMI-MLS tropospheric ozone column is available from 60°S to 60°N. The overall ozone patterns are qualitatively very similar in both datasets. In particular, the enhanced values of tropospheric ozone are very close for SUNLIT and OMI-MLS datasets, while the low TrOC





in the tropics over Indonesia is ~5 DU smaller in the SUNLIT dataset than in OMI-MLS. Note that the main SUNLIT TrOC dataset, which we discussed in Sect. 4.5, spans over the altitude range from the ground to the 3 km below the tropopause. The availability of gridded interpolated ozone profiles from limb sensors allows estimating the tropospheric ozone column with any upper limit, as it is done for the comparison with the OMI-MLS data.

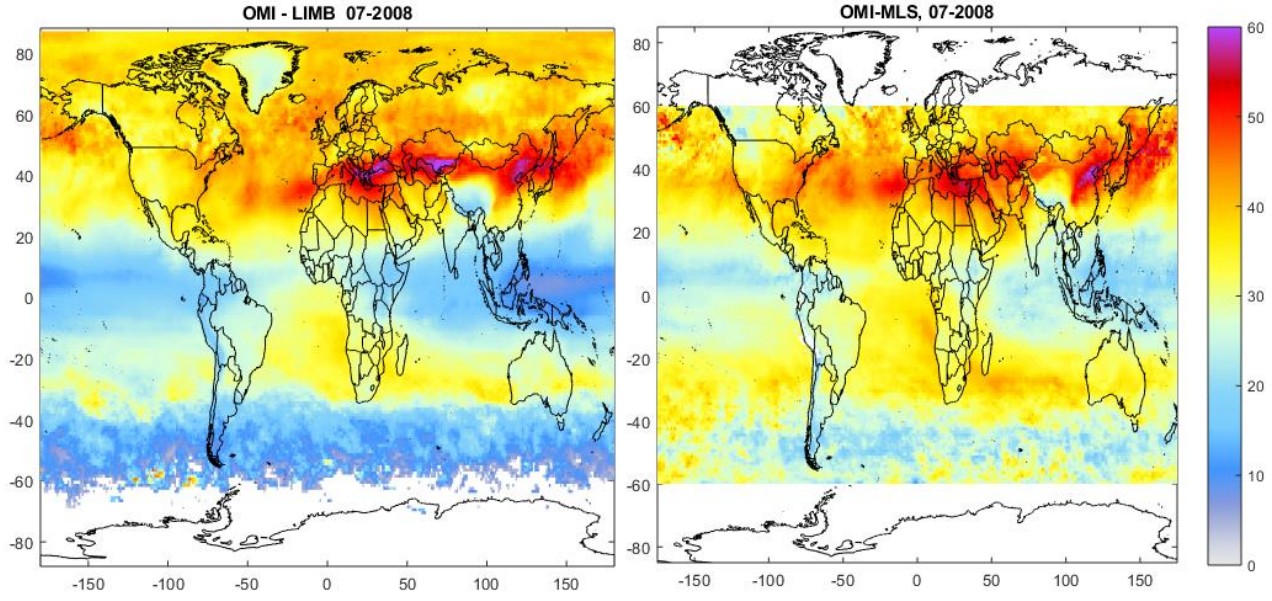


**Figure 15. Left: SUNLIT tropospheric ozone column for July 2008 (OMI minus limb SOC); right: NASA OMI-MLS tropospheric ozone column.**

## 5.3    Comparison with CCD ozone

The convective cloud differential method (CCD) allows retrievals of tropospheric ozone column in the tropical region, at latitudes 20°S-20°N. The CCD tropospheric ozone dataset has been developed in ozone CCI project; it represents the ozone column in the altitude range from ground to 10 km (Heue et al., 2016, https://climate.esa.int/en/projects/ozone/data/). For comparison with the CCD tropospheric ozone column, we integrated the limb ozone profiles from 10 km to 55 km, and subtracted it from clear-sky total ozone column. The comparison of tropospheric ozone columns from CCD method and from

our computations are presented in Figure 16.

The morphology of the ozone distribution in the tropics in September 2008 is similar in the OMI-CCD dataset and in our tropospheric ozone column taken from the ground up to 10 km. However, but the OMI-CCD TrOC values are ~5-7 DU higher than in our analysis (note different color scales on the Figure 16 panels). Heue et al. (2016) noticed a slightly larger ~1.7 DU) tropospheric ozone from the CCD method than in the collocated ozone-sonde values. Other differences can be due to using





several nadir sensor data in the CCD dataset (OMI and GOME-2 data are debiased to the SCIAMACHY dataset), as well as other approximations used in the processing of the CCD dataset (Heue et al., 2016).

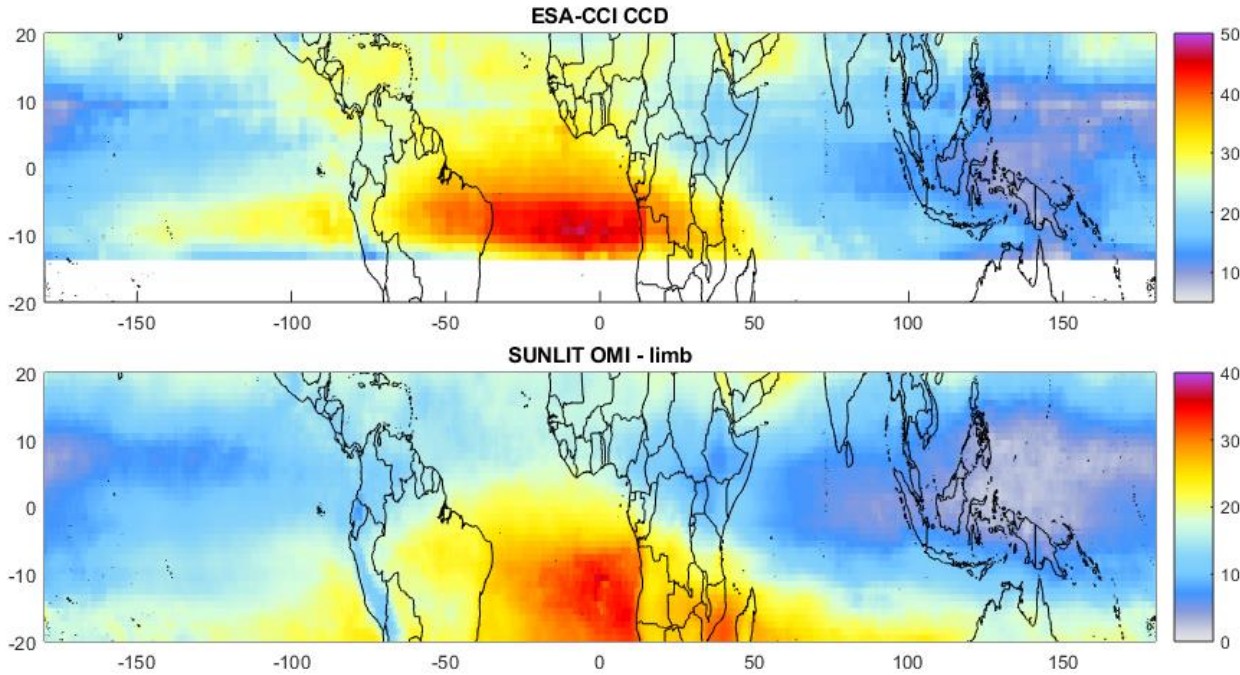

**Figure 16. Tropospheric ozone column in DU (color) for September 2008 from OMI CCD (top) and SUNLIT data (SOC is integrated from 10 km).**


## 6    Summary

In this paper, we have presented the results of our studies on the methods for retrievals of tropospheric ozone column by the residual method, i.e., the combination of total ozone column from nadir instruments with the stratospheric ozone column

from limb instruments. The main result of our studies, which are performed in the framework of the ESA SUNLIT project, is the tropospheric ozone column datasets obtained by combining the OMI and TROPOMI total ozone columns with ozone profiles from the limb satellite instruments. The data are the monthly-averaged distributions with the horizontal resolution of 1°x1°.

Other datasets, which are created as an intermediate step of creating the tropospheric ozone column data can be used in

other applications. These datasets are daily gridded with 1°x1° horizontal resolution and include (i) homogenized and



interpolated dataset of ozone profiles from limb instruments, (ii) stratospheric ozone column from limb instruments, and (iii) clear-sky and total ozone columns from nadir instruments.

The methodological developments made in our work include the method for homogenization of data from various satellite instruments and the method for horizontal interpolation, which takes into account both data uncertainties and variability of the parameter of interest.

The developed ozone datasets are in good agreement with ozone sonde and other satellite data. The global distributions of tropospheric ozone show clearly the enhancements associated with the regions of enhanced ozone production in the troposphere. The SUNLIT tropospheric ozone column dataset can be used in different analyses, including evaluation of long-term changes in the tropospheric ozone. This will be the subject of our work in the future.


**Data Availability**

The tropospheric and stratospheric ozone column data are in open access at Sodankylä National Satellite Data centre https://nsdc.fmi.fi/data/data_sunlit.php. The data cover the observational period from the beginning of the OMI and TROPOMI missions until December 2020. The daily 1°x1° homogenized dataset of ozone profiles can be obtained by request
from the first author.

**Author contributions**

VS is the PI of the SUNLIT project, the developer of the SUNLIT algorithms, and the writer of the major part of the manuscript. RH and MS provided SILAM simulations and participated in the feasibility studies. HSL participated in data processing,
validation, and analyses of tropospheric ozone. MSz participated in feasibility studies and data analyses. All authors (VS, RH, MS, MSz, HSL, JI and CR) participated in discussion of the algorithm and contributed to writing the paper

**Competing interests**

The authors declare that they have no conflict of interest.


**Acknowledgements**

The work is performed in the framework of the ESA project SUNLIT. The harmonized dataset of ozone profiles (HARMOZ) in created in the framework of ESA projects Ozone_cci and Ozone_cci+. SILAM model updates were supported by the GLORIA project of Academy of Finland (grant 310372). The authors thank the University of Bremen team for SCIAMACHY
data and HARMOZ-MLS data, the Karlsruhe Institute of Technology team for MIPAS data, the University Saskatchewan team for providing OSIRIS and OMPS-LP data, and the NASA/JPL team for providing MLS data.

**Financial support**



This research has been supported by the European Space Agency projects SUNLIT and Ozone_cci+, and the Academy of
Finland, Centre of Excellence of Inverse Modelling and Imaging (decision 336798).

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
