# Peer review of "Synergy of Using Nadir and Limb Instruments for Tropospheric Ozone Monitoring"

_Atmospheric Measurement Techniques, 2021_

## Referee Comment (RC1)

**Referee Report to "Synergy of Using Nadir and Limb Instruments for Tropospheric Ozone Monitoring" by Viktoria F. Sofieva et al.**

The manuscript describes a new approach to estimate tropospheric ozone column in the framework of the residual method by using multiple data sets from limb-viewing instruments to calculate the stratospheric ozone column. Methods to homogenize and interpolate the data are described. The suggested approach is certainly of interest for the scientific community and, in general, the study is of a good quality suitable for publication in AMT. However, there are a few deficits in the study that need to be addressed before the publication. These are mainly insufficient justifications of the approaches and obtained results. My detailed comments are listed below.

**Major comments**

- Sect. 3: I do not fully agree with the concept of the correction of the upper tropospheric ozone column using the data from an external source used by the authors to remove the UT contribution. From the point of view of atmospheric dynamics, I'd expect that the main source of the UT ozone is its transport form the lower troposphere. If the authors have a different opinion they have to provide and justify it. If the UT ozone is determined by the tropospheric pollution, it should be closely related to the ground sources and transport processes. Thus, the described correction can only work properly if the dominating ground sources do not change their strength an location. To my opinion this method can result in artifacts if distribution of the ground sources changes. With that it is not clear what will be the goal of the corrected data. Please provide more discussion/justification in the paper.
- The superiority of the interpolation approach over the data assimilation is stated but, in my opinion, not well justified.
- Supplement 3 is meant to demonstrate a good agreement of the small-scale ozone variability in OMI and SILAM data. Looking at Figs. 8-10 I cannot follow how the authors come to the conclusion that the agreement between the modeled and experimental data is very good, e.g. I see nothing in common between black or between red curves for 60°S 90°S in Figs. 9 and 10. As this part is not highly relevant for the rest of the study this supplement can be removed. Otherwise comparisons and justification of the conclusions must be improved.
- The provided comparisons for the tropospheric ozone are too sparse. Plots illustrating time series need also be provided (preferably as 2D plots rather contours as the latter are much more difficult to compare). The provided comparison illustrates that SUNLIT results are somewhat different from other data but no attempts is made to investigate, which dataset should be considered as a better one. Comparisons with ozonesondes for the resulting tropospheric ozone values (preferably including results from other datasets) are clearly missing and have to be added.

**Minor comments**

- In Abstract time ranges of the created data sets should be mentioned
- Page 2, lines 50-52: this is only true for the along line of sight direction. The resolution can be much higher in the across direction, e.g. ALTIUS, CAIRT.
- Page 2, line 54: Presence of clouds is also a problem for the nadir measurements and for the usage of the residual method in general.
- Page 2, line 63: Please add "Leventidou, E., Eichmann, K.-U., Weber, M., and Burrows, J. P.: Tropical tropospheric ozone columns from nadir retrievals of GOME-1/ERS-2, SCIAMACHY/Envisat, and GOME-2/MetOp-A (1996-2012), Atmos. Meas. Tech., 9, 3407-3427, https://doi.org/10.5194/amt-9-3407-2016, 2016." to the citations
- Page 3, line 71: The data calibration is not a serious issue then combining total/stratospheric ozone columns retrieved with DOAS-like methods
- Sect. 3.1: It is not clear why the UTLS region is treated separately, as UT is the inherent part of the troposphere and contributes to the tropospheric ozone while LS is a part of the stratosphere.
- Page 8, paragraph starting at line 200: It is incorrect to talk about UTLS here as you only consider the region below the tropopause and since do not enter the lower stratosphere (LS).
- Page 8, Sect. 3.2: I am wondering if the observed large influence of the UT region is specific to the selected method to determine the tropopause. Do the conclusions remain the same if using blended tropopause?
- Page 8, Sect. 3.2: The name of the section might be sub-optimal as one expects rather a discussion about integration effects. Here, "vertical extent" would be more appropriate.
- Page 9, line 223: The sentence duplicates information already presented in the two sentences above.
- Page 9, reference to Fig. S2: information needs to be given how the sampling for this plot was implemented, e.g. in accordance to which instrument's sampling pattern.
- The same comment as above applies to Figure 4.
- Page 9, Figure S3: Altitude axis in km should be provided in addition.

- Page 10, second item: "the observed ground-level ozone enhancements" is an incorrect formulation. As follows from the previous discussion, the ground-level ozone enhancements are almost not seen by the instruments. The increased ozone amounts become detectable when air masses raise over the boundary layer.
- Page 10, third item: this conclusion depends certainly on the sampling of the considered instruments and should not be stated in general. By the way, are the authors aware of any more or less recent publication where the residual method was applied to the monthly mean values? Isn't the recommendation not to combine the monthly mean values too obvious for the scientific community for now? Another point to this topic, as shown in Fig. S4 of the paper, there is quite a strong difference between the tropospheric ozone values calculated from daily means and from the collocated data. Thus, the recommendation given by authors to use the daily measurements can be confusing for the readers forcing them to prefer daily means to the fully collocated measurements.
- Page 11, line 271: please provide a reference discussing the OMI row anomaly
- Page 11, lines 274: "region between the two ozone jumps is removed" from the text above it is unclear which two ozone jumps are meant.
- Page 11, Figure 5: it is not quite clear if the plotted "random uncertainties" are the same as the "uncertainty of the total ozone column" given by Eq. (1)
- Page 12, lines 289 293: The logic of these two sentences is not clear. It is unclear how the described procedure "we first create the 1°x1° gridded and interpolated dataset of ozone profiles, and then we compute stratospheric column via integration of ozone profiles" can mitigate the issue that "the limb instruments have limited accuracy and highly non-uniform coverage in the UTLS". I can imaging that using multiple instruments might reduce the non-uniform coverage but I doubt it can significantly increase the accuracy. Please comment on that.
- Figure 7: I do not see any stars in the plot.
- Figure S5: It is not quite clear if the differences were interpolated or these are the differences between interpolated MLS and SILAM. Interpolation rule should be reported.
- Figure S6: It is not quite clear why a data assimilation should result in an artificial trend. Could you add the third panel to the figure showing the trends in the assimilated data? Otherwise, the conclusion about a disadvantage of the assimilation looks poorly justified.
- Sect. 4.3.3: The first sentence is misleading as it refers to the region below the tropopause. First, the profile values below the tropopause are not of interest as they are not accounted for when calculating the stratospherical ozone column. Second,

as stated below, the correction is made between two fixed pressure levels having no relation to the real tropopause hight.

- Figure 9: 200 hPa and 400 HPa levels should be marked in the plots.
- Figure S12: Sub-optimal color scale. It is almost impossible to estimate the plotted differences. The colors between 10 and 20 DU are almost indistinguishable.
- Page 17, line 394: Please comment on values over Southern America and Africa which seem to be around 10 DU or even larger.
- Page 17, lines 402-403: The correction of 2 DU is quite small and do not significantly change the results, the application of this correction is, however, questionable. This difference can result from an uncertainty in cloud top height definition or from the fact that the clouds are not a purely reflecting layer and radiation penetrates into the cloud to a certain depth. Thus, there might me a physical difference between the integrated limb profiles and total ozone observation in a cloudy atmosphere, which however is not applicable to cloud-free conditions. The correction should be either removed or better justified.

**Technical corrections**

- Page 1, line 9 (and also Page 2, line 48): "The satellite measurements"  $\rightarrow$  "Satellite measurements"
- Page 1, line 11: "total ozone column"  $\rightarrow$  "total ozone columns"
- Page 1, line 12: "stratospheric ozone column dataset"  $\rightarrow$  "stratospheric ozone column datasets"
- Page 1, lines 14-15: please reword the sentence to avoid a double usage of the word "using"
- Page 8, line 201: extra or missing bracket in "Figure 3, right panels)"
- Page 23, line 477: "However, but the OMI-CCD"  $\rightarrow$  "However, the OMI-CCD"

---

## Author Comment (AC1)

Dear Reviewer,

Thank you very much for your comments on our manuscript. We took your comments into account in the revised version of the manuscript. Please find below our detailed replies (black font) on your comments (blue font).

General comments:

This manuscript introduces a new methodology aiming to infer global distributions of tropospheric ozone columns making use of the synergy between nadir and limb-viewing satellite ozone observations. Although the principle of the residual method used is well established and was applied in a number of other studies, the approach proposed here is innovative because it is based on the combination of several limb-viewing satellite instruments to infer the stratospheric column reference. This greatly expands the potential of applicability of the method and may lead (in the future) to the generation of long-time series suitable for trend evaluations. In this study however, the authors concentrates on a demonstration of the concept. Difficulties inherent to the residual technique are investigated in details using 3D CTM simulations by the SILAM model and these simulations are used to design a retrieval approach that mitigates at best the main sources of uncertainty. Reading through the manuscript is very instructive and leaves the reader with a better understanding of the information content of the technique and its limitations. In particular, I found the approach used to deal with uncertainties quite robust and convincing. Practical applications are limited to a few example based on OMI and TROPOMI data and therefore it remains to evaluate whether the proposed approach will be stable over time, especially when different limb-sensors have to be combined. The validation effort concentrates on an assessment of the homogenised stratospheric ozone profiles data generated as an input to the algorithm, as well as a few comparisons of the resulting tropospheric ozone product with alternative products (OMI-MLS and CCD). At this point, one may wonder: why not attempting to also validate the tropospheric ozone product with ozone sonde data? Since a data base of ozone sonde measurements was already assembled to validate stratospheric profiles, it seems to me that it could also be used for comparison with the resulting tropospheric ozone columns.

In the revised version, we added a figure with comparison of time series of tropospheric ozone column from SUNLIT (OMI-LIMB) and from integrated ozonesonde profiles, at locations of several ozonesonde stations. We compare also seasonal cycle of tropospheric ozone derived from these SUNLIT and ozonesonde data. We added also a discussion on differences in sampling pattern of ozonesonde and satellite data.

Anyway, I found this study very interesting and promising and I look forward to see the method applied on a larger scale. The English writing however is not always up to AMT standards and I strongly recommend that authors get help from a native English speaker to polish their text.

All Copernicus publications are proofread by a native speaker at the final stage.

Other than that, the manuscript is clear overall, figures are of good quality and adequate in number and references give a good credit to the published literature on the subject. I therefore recommend publication in AMT, after attention to the few comments and suggestions below.

Detailed comments:

Pg. 1, l. 14: replace 'using' (at the end of the line) by 'supported by'

Corrected.

Pg. 2, l. 42: this sentence does not read well. 'Ozone' is not a concern as such, but the impact of its changes on human health, climate change, etc are clear environmental concerns.

Rephrased as suggested

Pg. 2, l. 55: 'an effective combination of the limb and nadir measurements … can provide a new information'. Here I would rather say 'provide additional information'

Corrected

Pg. 3, l. 63: add a reference to Heue et al., 2016, after Ziemke et al., 1998. This reference is already in your list.

The reference is added.

Pg. 3, l. 90: I think that it would be useful to already mention in the introduction that the methods being developed in the study have a focus on optimizing monthly-averaged tropospheric ozone values, which are mostly interesting for long-term studies and climatological analysis.

Thank you, we added this note in the introduction.

Pg. 4, l. 100: mention here that the GODFIT v4.0 processor was developed as part of the ESA Ozone_cci project (like you do in Pg. 5 for the HARMOZ data).

We mention this in the revised version.

Pg. 5, Fig. 1:  there seems to be some mismatch in the instrument's labels. I suppose that ACE should be replaced by OMPS-LP. Also one curve seems to be missing (only 5 curves are displayed while Table 1 refers to 6 instruments). Again, I suppose that OMPS-LP is the missing one.

The instruments labels are indeed incorrect, there should not be "ACE-FTS". Since the limb instruments operated in different years (and there is no period when all 6 instruments operated), we selected year 2008, when data from 5 out of 6 instruments are available. That is why OMPS-LP is not present in Fig.1. We corrected Figure 1

Pg. 7, l. 172: you might add a reference discussing the chemical links between tropospheric ozone and its precursors (NOx and VOCs).

We added the reference: Seinfeld, J.H., Pandis, S.N., 2006. Atmospheric chemistry and physics: from air pollution to climate change, 2nd ed., ed. J. Wiley, Hoboken, N.J., chapter on Tropospheric Chemistry.

Pg. 7, l. 186: add 'gradient' between 'concentration' and 'drops'

Corrected

Pg. 7, Fig. 2: add a name or short description for the different layers considered on the figure

We will add names of layers in Figure 2

Pg. 9, l. 225: indicate which limb satellite instrument was used to prescribe the sampling applied in Figure S2. Is it MLS only or the combined data set of MLS, OSIRIS and OMPS instruments (as shown in Fig. 7). Same comment regarding Fig. 4.

The sampling patterns correspond to the combined datasets. In the revised version, we name explicitly the instruments used in the combined datasets in Figure S2 and Fig.4

Pg. 11, l. 271: the OMI row anomaly is currently not introduced in the manuscript. Please add a reference or better describe the nature of the problem.

We added the reference (Schenkeveld et al., 2017).

Pg. 22, Fig. 14: replace 'pressure altitude' by 'altitude' as legend for the right y-axis

The right vertical axis is "pressure altitude", not geometric altitude.

Spelling, typos:

Pg. 1, l. 1-2: avoid repetition of the word 'provide'

Pg. 3, l. 80: remove 'the' between 'using' and 'simulations'

Pg. 4, l. 107: remove 'In our work' (to avoid repetition with the previous paragraph)

Pg. 7, l. 173: remove 'the' between 'at' and 'altitudes'

Pg. 7, l. 176: remove 'the' between 'from' and 'fluctuations'

Pg. 7, l. 182: … the model data 'are' either used in their entirety or sub-sampled at 'the' location and times of…

Pg. 9, l. 210: add 'the' between 'consider' and 'possibility'

Pg. 9, l. 221: add 'of' between 'averaging' and 'data'

Pg. 10, l. 256: …we have developed 'a' method of estimating…

Pg. 11, l. 273: remove 'the' between 'If' and 'values'

Pg. 12, l. 295: correct 'horizonal' by 'horizontal'

Pg. 14, l. 320: remove 'the' between 'provides' and 'random'

Pg. 14, l. 328: remove 'the' between 'By' and 'construction'

Pg. 15, l. 347: replace 'The example…' by 'An example…'

Pg. 22, l. 456: remove 'the' between 'dataset' and 'examples'

Pg. 23, l. 461: add 'However' at the beginning of the sentence starting with 'The availability of gridded interpolated ozone profiles…'

Pg. 23, 471: add 'the' between 'in' and 'ozone CCI'

Pg. 23, l. 477, remove 'but' after 'However'

All are corrected, thank you.

---

## Author Comment (AC2)

Dear Reviewer,

Thank you very much for very careful reading our manuscript and for your comments. We took your comments into account in the revised version of the manuscript. Please find below our detailed replies (black font) on your comments (blue font).

The manuscript describes a new approach to estimate tropospheric ozone column in the framework of the residual method by using multiple data sets from limb-viewing instruments to calculate the stratospheric ozone column. Methods to homogenize and interpolate the data are described. The suggested approach is certainly of interest for the scientific community and, in general, the study is of a good quality suitable for publication in AMT.
However, there are a few deficits in the study that need to be addressed before the publication. These are mainly insufficient justifications of the approaches and obtained results.
My detailed comments are listed below.

**Major comments**

1. Sect. 3: I do not fully agree with the concept of the correction of the upper tropospheric ozone column using the data from an external source used by the authors to remove the UT contribution. From the point of view of atmospheric dynamics, I'd expect that the main source of the UT ozone is its transport form the lower troposphere. If the authors have a different opinion they have to provide and justify it. If the UT ozone is determined by the tropospheric pollution, it should be closely related to the ground sources and transport processes. Thus, the described correction can only work properly if the dominating ground sources do not change their strength and location. To my opinion this method can result in artifacts if distribution of the ground sources changes. With that it is not clear what will be the goal of the corrected data. Please provide more discussion/justification in the paper.

According to (Škerlak et al., 2014; Young et al., 2018), stratosphere-troposphere exchange has an important role in the upper tropospheric ozone budget (several tens of percent).
The upper tropospheric correction described in Sect. 3 in our feasibility study is a very approximate one (as any correction by climatological values), and will suffer from deficiencies, which you noted. That is why in our SUNLIT processing we use the UTLS profiles from the model adjusted to measurements. In the revised version, we added the caveats in Section 3.

• The superiority of the interpolation approach over the data assimilation is stated but, in my opinion, not well justified.

In the revised version, we added references ( e.g., Simmons et al., 2014; Stauffer et al., 2019), which discuss the problems of using assimilated data for trend analyses (see also a more detailed reply below).

• Supplement 3 is meant to demonstrate a good agreement of the small-scale ozone variability in OMI and SILAM data. Looking at Figs. 8-10 I cannot follow how the authors come to the conclusion that the agreement between the modeled and experimental data is very good, e.g. I see nothing in common between black or between red curves for 60_S { 90_S in Figs. 9 and 10. As this part is not highly relevant for the rest of the study this supplement can be removed. Otherwise comparisons and justification of the conclusions must be improved.

Yes, the disagreement, which you note, is indeed observed for the band 60-90S June-Aug and Sep-Nov. We would like to note that OMI cannot measure in polar night conditions, therefore such disagreement is expected due to limited OMI coverage in these seasons and locations (the same is valid also for the NH). For stratospheric ozone column, we use only cloudy pixels of OMI, which have limited coverage. Additional disagreement comes from biases between model and observations. For the SUNLIT processing, model biases are not important, since we use the adjusted model field. We note this in the revised version of the Supplement.

• The provided comparisons for the tropospheric ozone are too sparse. Plots illustrating time series need also be provided (preferably as 2D plots rather contours as the latter are much more difficult to compare). The provided comparison illustrates that SUNLIT results are somewhat different from other data but no attempts is made to investigate, which dataset should be considered as a better one. Comparisons with ozonesondes for the resulting tropospheric ozone values (preferably including results from other datasets) are clearly missing and have to be added.

In the revised version, we added a figure with comparison of time series of tropospheric ozone column from SUNLIT (OMI-LIMB) and from integrated ozonesonde profiles, at locations of several ozonesonde stations. We compare also seasonal cycle of tropospheric ozone derived from these SUNLIT and ozonesonde data. A good agreement is observed.

We added also a discussion on differences in sampling pattern of ozonesonde and satellite data. We added also a note on ongoing TOAR-II activity aimed at comparison of different tropospheric ozone columns, and efforts on making different tropospheric datasets compatible for comparison.

**Minor comments**
• In Abstract time ranges of the created data sets should be mentioned

In the revised abstract, we added: "The datasets are processed from the beginning on OMI and TROPOMI measurements until Dec 2020, and they will be regularly extended in future".

• Page 2, lines 50-52: this is only true for the along line of sight direction. The resolution can be much higher in the across direction, e.g. ALTIUS, CAIRT.

We added "along line of sight" in the revised version.

• Page 2, line 54: Presence of clouds is also a problem for the nadir measurements and for the usage of the residual method in general.

We agree and added this note.

• Page 2, line 63: Please add \Leventidou, E., Eichmann, K.-U., Weber, M., and Burrows, J. P.: Tropical tropospheric ozone columns from nadir retrievals of GOME- 1/ERS-2, SCIAMACHY/Envisat, and GOME-2/MetOp-A (1996-2012), Atmos. Meas. Tech., 9, 3407-3427, https://doi.org/10.5194/amt-9-3407-2016, 2016." to the citations

The reference is added.

• Page 3, line 71: The data calibration is not a serious issue then combining total/stratospheric ozone columns retrieved with DOAS-like methods

We agree the calibration is not a serious issue, but still an issue (Fishman and Larsen, 1987).

• Sect. 3.1: It is not clear why the UTLS region is treated separately, as UT is the inherent part of the troposphere and contributes to the tropospheric ozone while LS is a part of the stratosphere.

Some studies define the tropospheric ozone until the tropopause, some studies exclude the uppermost troposphere. In the revised version, the atmospheric layers are named in Figure 2, so that the readers will see clearly their contribution.

• Page 8, paragraph starting at line 200: It is incorrect to talk about UTLS here as you only consider the region below the tropopause and since do not enter the lower stratosphere (LS).

Yes, it should be "upper troposphere", corrected.

• Page 8, Sect. 3.2: I am wondering if the observed large influence of the UT region is specific to the selected method to determine the tropopause. Do the conclusions remain the same if using blended tropopause?

The conclusions will remain the same also for blended tropopause (or dynamical tropopause). This can be seen clearly in the tropics, for example, where blended and thermal tropopause are very close/coincide.

• Page 8, Sect. 3.2: The name of the section might be sub-optimal as one expects rather a discussion about integration effects. Here, \vertical extent" would be more appropriate.

Changed as suggested.

• Page 9, reference to Fig. S2: information needs to be given how the sampling for this plot was implemented, e.g. in accordance to which instrument's sampling pattern.
• The same comment as above applies to Figure 4.

In the revised version, we name explicitly the instruments used in the combined datasets in Figure S2 and Fig.4

• Page 9, Figure S3: Altitude axis in km should be provided in addition.

We will add the approximate altitude axis.

• Page 10, second item: "the observed ground-level ozone enhancements" is an incorrect formulation. As follows from the previous discussion, the ground-level ozone enhancements are almost not seen by the instruments. The increased ozone amounts become detectable when air masses raise over the boundary layer.

"Ground-level" words were redundant and they are removed from this sentence.

• Page 10, third item: this conclusion depends certainly on the sampling of the considered instruments and should not be stated in general. By the way, are the authors aware of any more or less recent publication where the residual method was applied to the monthly mean values? Isn't the recommendation not to combine the monthly mean values too obvious for the scientific community for now? Another point to this topic, as shown in Fig. S4 of the paper, there is quite a strong difference between the tropospheric ozone values calculated from daily means and from the collocated data. Thus, the recommendation given by authors to use the daily measurements can be confusing for the readers forcing them to prefer daily means to the fully collocated measurements.

Although we are now aware about recent publications on the residual method applied to monthly mean values, we think it is worth to keep this statement. We agree that it is rather obvious, and added "obviously" to this sentence.
It is noted in the paper that SUNLIT tropospheric ozone column correspond to the local time of OMI and TROPOMI measurements, not daily mean. We will stress this more in the revised version.

• Page 11, line 271: please provide a reference discussing the OMI row anomaly

We added the reference (Schenkeveld et al., 2017).

• Page 11, lines 274: "region between the two ozone jumps is removed" - from the text above it is unclear which two ozone jumps are meant.

We changes "ozone jumps" to "pixels with huge ozone gradient".

• Page 11, Figure 5: it is not quite clear if the plotted "random uncertainties" are the same as the "uncertainty of the total ozone column" given by Eq. (1)

Yes, these are the same and we indicate this in the revised version.

• Page 12, lines 289 - 293: The logic of these two sentences is not clear. It is unclear how the described procedure "we first create the 1_x1_ gridded and interpolated dataset of ozone profiles, and then we compute stratospheric column via integration of ozone profiles" can mitigate the issue that "the limb instruments have limited accuracy and highly non-uniform coverage in the UTLS". I can imaging that using multiple instruments might reduce the non-uniform coverage but I doubt it can significantly increase the accuracy. Please comment on that.

The advantage of this approach is discussed below in the text. The multiple instruments reduce non-uniform coverage, thus reducing interpolation errors. In our approach, we use adjusted SILAM model in the upper troposphere, which allows a significant improvement of ozone profiles data in the UTLS. These aspects are discussed below in our paper, therefore we added "see details below" to these sentences.

• Figure 7: I do not see any stars in the plot.

[Figure]

Please look at the marked by oval area. They are also in other locations in NH.

• Figure S5: It is not quite clear if the differences were interpolated or these are the differences between interpolated MLS and SILAM. Interpolation rule should be reported.

As stated in the text, this is "the interpolated absolute difference between MLS and SILAM adjusted data". In the revised version, we indicate the interpolation rule.

• Figure S6: It is not quite clear why a data assimilation should result in an artificial trend. Could you add the third panel to the figure showing the trends in the assimilated data? Otherwise, the conclusion about a disadvantage of the assimilation looks poorly justified.

The problems of using the assimilated data for trend analyses are well documented in the literature. Inhomogeneities and discontinuities can be introduced by a changing number of assimilated datasets over time. In the revised version, we added references ( e.g., Simmons et al., 2014; Stauffer et al., 2019), which discuss these issues.

• Sect. 4.3.3: The first sentence is misleading as it refers to the region below the tropopause. First, the profile values below the tropopause are not of interest as they are not accounted for when calculating the stratospherical ozone column. Second, as stated below, the correction is made between two fixed pressure levels having no relation to the real tropopause height.

The values below the tropopause are of interest. For some applications – and also for our dataset – the stratospheric column includes the UTLS region.

• Figure 9: 200 hPa and 400 HPa levels should be marked in the plots.

We will mark these levels in the revised version.

• Figure S12: Sub-optimal color scale. It is almost impossible to estimate the plotted differences. The colors between 10 and 20 DU are almost indistinguishable.

We will improve the color representation.

• Page 17, line 394: Please comment on values over Southern America and Africa which seem to be around 10 DU or even larger.
• Page 17, lines 402-403: The correction of 2 DU is quite small and do not significantly change the results, the application of this correction is, however, questionable. This difference can result from an uncertainty in cloud top height definition or from the fact that the clouds are not a purely reflecting layer and radiation penetrates into the cloud to a certain depth. Thus, there might me a physical difference between the integrated limb profiles and total ozone observation in a cloudy atmosphere, which however is not applicable to cloud-free conditions. The correction should be either removed or better justified.

In the revised version, we added that this correction can be further tuned in future, when extensive validation of tropospheric ozone column data will be performed.

**Technical corrections**
• Page 1, line 9 (and also Page 2, line 48): "The satellite measurements" -> "Satellite measurements"
• Page 1, line 11: "total ozone column" -> "total ozone columns"
• Page 1, line 12: "stratospheric ozone column dataset" -> "stratospheric ozone column datasets"
• Page 1, lines 14-15: please reword the sentence to avoid a double usage of the word "using"
• Page 8, line 201: extra or missing bracket in "Figure 3, right panels)"
• Page 23, line 477: "However, but the OMI-CCD" -> "However, the OMI-CCD"

All are corrected. Thank you.

REFERENCES

Fishman, J. and Larsen, J. C.: Distribution of total ozone and stratospheric ozone in the tropics: Implications for the distribution of tropospheric ozone, J. Geophys. Res. Atmos., 92(D6), 6627–6634, doi:10.1029/JD092iD06p06627, 1987.

Simmons, A. J., Poli, P., Dee, D. P., Berrisford, P., Hersbach, H., Kobayashi, S. and Peubey, C.: Estimating low-frequency variability and trends in atmospheric temperature using ERA-Interim, Q. J. R. Meteorol. Soc., 140(679), 329–353, doi:10.1002/qj.2317, 2014.

Škerlak, B., Sprenger, M. and Wernli, H.: A global climatology of stratosphere–troposphere exchange using the ERA-Interim data set from 1979 to 2011, Atmos. Chem. Phys., 14(2), 913–937, doi:10.5194/acp-14-913-2014, 2014.

Stauffer, R. M., Thompson, A. M., Oman, L. D. and Strahan, S. E.: The Effects of a 1998 Observing System Change on MERRA-2-Based Ozone Profile Simulations, J. Geophys. Res. Atmos., 124(13), 7429–7441, doi:https://doi.org/10.1029/2019JD030257, 2019.

Young, P. J., Naik, V., Fiore, A. M., Gaudel, A., Guo, J., Lin, M. Y., Neu, J. L., Parrish, D. D., Rieder, H. E., Schnell, J. L., Tilmes, S., Wild, O., Zhang, L., Ziemke, J., Brandt, J., Delcloo, A., Doherty, R. M., Geels, C., Hegglin, M. I., Hu, L., Im, U., Kumar, R., Luhar, A., Murray, L., Plummer, D., Rodriguez, J., Saiz-Lopez, A., Schultz, M. G., Woodhouse, M. T. and Zeng, G.: Tropospheric Ozone Assessment Report: Assessment of global-scale model performance for global and regional ozone distributions, variability, and trends, edited by D. Helmig and A. Lewis, Elem. Sci. Anthr., 6, 10, doi:10.1525/elementa.265, 2018.

---

## Referee Report (RR1)

**Referee report to the revised version of "Synergy of Using Nadir and Limb Instruments for Tropospheric Ozone Monitoring" by Viktoria F. Sofieva et al.**

The revised manuscript has been improved. However, some of my comments have not been addressed to the full extend. The manuscript would benefit from a few additional minor changes. My detailed comments are listed below.

For minor comments, my original comments are written in blue, the responses of the authors in green and my additional comments in black.

**Major comments**

My comment "Supplement 3 is meant to demonstrate a good agreement of the small-scale ozone variability in OMI and SILAM data. Looking at Figs. 8-10 I cannot follow how the authors come to the conclusion that the agreement between the modeled and experimental data is very good, e.g. I see nothing in common between black or between red curves for 60°S – 90°S in Figs. 9 and 10. As this part is not highly relevant for the rest of the study this supplement can be removed. Otherwise comparisons and justification of the conclusions must be improved." was addressed only in part. The strong disagreement at high southern latitudes is properly explained and the remaining agreement is claimed to be good. However, one still sees a lot of discrepancies, e.g. in Fig. 9 and 10 for 30°S – 30°N region the behavior of most of the curves is quite different for distances larger than 500 km, for 30°N – 60°N the blue, green, black and red solid curves are grouped clearly different, for 30°S – 60°S red and black dashed curves do not look similar. For this reason I do not agree with rating the agreement as good without any additional comments.

**Minor comments**

"Page 2, lines 50-52: this is only true for the along line of sight direction. The resolution can be much higher in the across direction, e.g. ALTIUS, CAIRT."

"We added 'along line of sight' in the revised version."

This must be a misunderstanding. I meant that your sentence is only true for the along track horizontal resolution while the across track horizontal resolution can be much higher (forward/backward view is assumed). With that the correction in the revised version of the manuscript is not appropriate. Suggestion: "The measurements in the limb-viewing geometry have usually a good vertical resolution but their horizontal resolution is limited by the spatial sampling. In particular, the horizontal resolution in the along line of sight direction is limited by the effective horizontal length of interaction with the atmosphere (a few hundreds of kilometers)."

Page 3, line 71: The data calibration is not a serious issue then combining total/stratospheric ozone columns retrieved with DOAS-like methods

"We agree the calibration is not a serious issue, but still an issue (Fishman and Larsen, 1987)."

As far as I know both TOMS and SAGE algorithms discussed by (Fishman and Larsen, 1987) are not DOAS-like methods. Suggestion: " Aside with calibration issues when using a combination of TOMS and SAGE instruments, there was also..."

"Page 10, third item: this conclusion depends certainly on the sampling of the considered instruments and should not be stated in general. By the way, are the authors aware of any more or less recent publication where the residual method was applied to the monthly mean values? Isn't the recommendation not to combine the monthly mean values too obvious for the scientific community for now? Another point to this topic, as shown in Fig. S4 of the paper, there is quite a strong difference between the tropospheric ozone values calculated from daily means and from the collocated data. Thus, the recommendation given by authors to use the daily measurements can be confusing for the readers forcing them to prefer daily means to the fully collocated measurements."

"Although we are now aware about recent publications on the residual method applied to monthly mean values, we think it is worth to keep this statement. We agree that it is rather obvious, and added 'obviously' to this sentence. It is noted in the paper that SUNLIT tropospheric ozone column correspond to the local time of OMI and TROPOMI measurements, not daily mean. We stress this more in the revised version."

I do not mind to keep the recommendation not to use monthly mean data to calculate the tropospheric columns. However, I am still against a recommendation to use daily values. The usage of collocated measurements should still be preferred if feasible.

Page 11, lines 274: "region between the two ozone jumps is removed" - from the text above it is unclear which two ozone jumps are meant.

We changes "ozone jumps" to "pixels with huge ozone gradient".

It is still unclear what the authors mean. Are the pixels with values over 100 DU and their neighboring pixels are removed? Does it apply to one or more neighboring pixel at each side?

---

## Author Response (AR2)

Dear Reviewer and Editor,

Thank you very much for your comments. Please find below our replies.

**Major comments**

My comment "Supplement 3 is meant to demonstrate a good agreement of the small-scale ozone variability in OMI and SILAM data. Looking at Figs. 8-10 I cannot follow how the authors come to the conclusion that the agreement between the modeled and experimental data is very good, e.g. I see nothing in common between black or between red curves for  $60^{\circ}S - 90^{\circ}S$  in Figs. 9 and 10. As this part is not highly relevant for the rest of the study this supplement can be removed. Otherwise comparisons and justification of the conclusions must be improved." was addressed only in part. The strong disagreement at high southern latitudes is properly explained and the remaining agreement is claimed to be good. However, one still sees a lot of discrepancies, e.g. in Fig. 9 and 10 for  $30^{\circ}S - 30^{\circ}N$  region the behavior of most of the curves is quite different for distances larger than 500 km, for  $30^{\circ}N - 60^{\circ}N$  the blue, green, black and red solid curves are grouped clearly different, for  $30^{\circ}S - 60^{\circ}S$  red and black dashed curves do not look similar. For this reason I do not agree with rating the agreement as good without any additional comments.

We wrote a more detailed discussion on comparison of structure function from model and OMI data. In the revised version, instead of using a subjective word "good", we explained in detail agreement and disagreement of experimental and modelled structure functions for total and stratospheric ozone columns. The text in the second revision is:

"The overall morphology - latitudinal dependence, latitude-longitude anisotropy, seasonal cycle - is similar for OMI and SILAM, for both total and stratospheric ozone column. . For total ozone column, the experimental and modelled structure functions are very similar for almost all latitudinal zones. Some disagreement in seasonal cycle is observed for polar winter conditions (for example 60-90S in June-Aug, Figs S7 and S8). This disagreement is quite expected: OMI cannot measure in polar night conditions. The shape of structure functions and the growth with separation distance are similar in Figs S7 and S8, but some difference in absolute values exists and is expected; it comes from biases between model and observations (note that the structure functions are presented in absolute values).

For stratospheric ozone column, the comparison is more complicated, because we could use only cloudy pixels of OMI, which have limited coverage. This results in less reliable estimates of structure function from the OMI data. For example, limited amount of data at large separations (> 500-1000 km), resulted in different shapes of experimental and model-based the structure functions in the equatorial zone. Although the seasonal cycle and latitude-longitude anisotropy are qualitatively similar in Figures S9 and S10, mid- and high-latitude structure functions tend to

group somewhat differently. Therefore, comparison of the stratospheric ozone column structure functions in Figs S9 and S10 should be considered as indicative only.

Since the stratospheric ozone has a bulk contribution to the total ozone (for which observational and modelled structure functions are similar), and with the above notes, we conclude that the ozone small-scale variability is realistically represented by SILAM."

**Minor comments**

"Page 2, lines 50-52: this is only true for the along line of sight direction. The resolution can be much higher in the across direction, e.g. ALTIUS, CAIRT."

"We added 'along line of sight' in the revised version."

This must be a misunderstanding. I meant that your sentence is only true for the along track horizontal resolution while the across track horizontal resolution can be much higher (forward/backward view is assumed). With that the correction in the revised version of the manuscript is not appropriate. Suggestion: "The measurements in the limb-viewing geometry have usually a good vertical resolution but their horizontal resolution is limited by the spatial sampling. In particular, the horizontal resolution in the along line of sight direction is limited by the effective horizontal length of interaction with the atmosphere (a few hundreds of kilometers)."

**Corrected as suggested.**

Page 3, line 71: The data calibration is not a serious issue then combining total/stratospheric ozone columns retrieved with DOAS-like methods

"We agree the calibration is not a serious issue, but still an issue (Fishman and Larsen, 1987)."

As far as I know both TOMS and SAGE algorithms discussed by (Fishman and Larsen, 1987) are not DOAS-like methods. Suggestion: "Aside with calibration issues when using a combination of TOMS and SAGE instruments, there was also..."

Corrected as suggested.

"Page 10, third item: this conclusion depends certainly on the sampling of the considered instruments and should not be stated in general. By the way, are the authors aware of any more or less recent publication where the residual method was applied to the monthly mean values? Isn't the recommendation not to combine the monthly mean values too obvious for the scientific community for now? Another point to this topic, as shown in Fig. S4 of the paper, there is quite a strong difference between the tropospheric ozone values calculated from daily means and from the collocated data. Thus, the recommendation given by authors to use the daily measurements can be confusing for the readers forcing them to prefer daily means to the fully collocated measurements."

"Although we are now aware about recent publications on the residual method applied to monthly mean values, we think it is worth to keep this statement. We agree that it is rather obvious, and added 'obviously' to this sentence. It is noted in the paper that SUNLIT tropospheric ozone column correspond to the local time of OMI and TROPOMI measurements, not daily mean. We stress this more in the revised version."

I do not mind to keep the recommendation not to use monthly mean data to calculate the tropospheric columns. However, I am still against a recommendation to use daily values. The usage of collocated measurements should still be preferred if feasible.

This seems to be a misunderstanding. We do not state that the daily values are preferable, compared to fully collocated data. To avoid the potential confusion here in the text, we changed the statement:

"Due to large variability of ozone field and limited sampling by satellite instruments, nadir and limb measurements should be collocated in time and space, if feasible."

Page 11, lines 274: "region between the two ozone jumps is removed" - from the text above it is unclear which two ozone jumps are meant.

We changes "ozone jumps" to "pixels with huge ozone gradient".

It is still unclear what the authors mean. Are the pixels with values over 100 DU and their neighboring pixels are removed? Does it apply to one or more neighboring pixel at each side?

We rephrased: "The presence of row anomaly was also checked by evaluating the ozone difference in neighbouring rows. Along the swath direction, the anomaly is visible as a sudden drop and rise of the retrieved ozone column. The procedure was checking a difference in neighbouring pixels; if larger than 100 DU drop and rise are detected, all pixels between these two points were removed."